# Zeta Inhibitory Peptide attenuates learning and memory by inducing NO-mediated downregulation of AMPA receptors

Alexey Bingor [1], Tomer Haham[2], Claire Thornton [3], Yael Stern-Bach [2] & Rami Yaka [1✉]

Zeta inhibitory peptide (ZIP), a PKMζ inhibitor, is widely used to interfere with the maintenance of acquired memories. ZIP is able to erase memory even in the absence of PKMζ, via an unknown mechanism. We found that ZIP induces redistribution of the AMPARGluA1 in HEK293 cells and primary cortical neurons, and decreases AMPAR-mediated currents in the nucleus accumbens (NAc). These effects were mimicked by free arginine or by a modified ZIP in which all but the arginine residues were replaced by alanine. Redistribution was blocked by a peptidase-resistant version of ZIP and by treatment with the nitric oxide (NO)-synthase inhibitor L-NAME. ZIP increased GluA1-S831 phosphorylation and ZIP-induced redistribution was blocked by nitrosyl-mutant GluA1-C875S or serine-mutant GluA1-S831A. Introducing the cleavable arginine-alanine peptide into the NAc attenuated expression of cocaine-conditioned reward. Together, these results suggest that ZIP may act as an arginine donor, facilitating NO-dependent downregulation of AMPARs, thereby attenuating learning and memory.

[1] Institute for Drug Research (IDR), School of Pharmacy, The Hebrew University of Jerusalem, Jerusalem 91120, Israel. [2] Institute for Medical Research Israel-Canada (IMRIC), Faculty of Medicine, The Hebrew University of Jerusalem, Jerusalem 91120, Israel. [3] Department of Comparative Biomedical Sciences, Royal Veterinary College, London, UK. ✉email: yaka@md.huji.ac.il

Long-term potentiation (LTP) is the leading cellular mechanism of learning and memory, and sustained activation of protein kinase M Zeta (PKMζ) is essential for its maintenance[1]. This concept mostly relies on the ability of Zeta Inhibitory Peptide (ZIP), originally synthesized as a pseudo-substrate inhibitor of PKMζ[2–4], to interfere with LTP and erase memory in a variety of learning tasks including spatial sensitization[5], pain sensitization[6], fear-associated memories[7], conditioned taste aversion[8] and development of locomotor sensitization in rodents[9]. Remarkably, however, two parallel studies generating transgenic mice lacking PKMζ found no impairments in learning and memory[10,11]. More surprisingly, application of ZIP or its scrambled PKMζ-inactive form (scrZIP) to hippocampal slices of the PKMζ knockout mice, abolished an established LTP[11]. It seems, therefore, that PKMζ is not required for LTP or for learning and memory, and that the effects of ZIP (and scrZIP) on these parameters must occur independently of PKMζ inhibition. The mechanism of ZIP action remains an open puzzling question.

Previously, we showed that microinjection of ZIP into the nucleus accumbens (NAc), a key structure within the reward system, abolished cocaine-conditioned reward[12], as also shown subsequently in PKMζ knockout mice[10]. In the current study, we examine the hypothesis that ionotropic glutamate AMPA receptors (AMPARs) may be a target of the action of ZIP. Therefore, we characterized the effect of ZIP on AMPARs using heterologous expression in HEK293 cells and primary cortical neurons, and electrophysiological recordings from NAc slices as well as behavioral studies.

Our findings suggest a molecular mechanism in which membrane-bound myristoylated ZIP may act as an arginine donor, thereby inducing nitrosylation-mediated endocytosis of AMPARs. Subsequent downregulation of AMPARs activity results in attenuation of cocaine-conditioned reward.

## Results

**ZIP induces GluA1 redistribution in HEK293 cells and neurons**. Since it was well established that ZIP reduces EPSCs in CA1 hippocampal neurons together with its ability to erase established memories in PKMζ knockout mice[10,11], we hypothesized that ZIP functionally interacts with AMPARs. To test this hypothesis, we transfected HEK293 cells with different AMPAR GFP-tagged subunits and visualized them with confocal fluorescence microscopy. When transfected HEK293 cells were exposed to ZIP (5 μM), we observed a time-dependent redistribution of GluA1 through the cell body (Fig. 1a, b). Similar results were observed with scrZIP (5 μM), but not with vehicle (PBS). This effect was AMPAR-subunit specific since bath application of ZIP or scrZIP to HEK293 expressing GFP-GluA2 had no effect on the distribution (Fig. 1c). To confirm that this observation was independent of PKMζ, we tested its expression in HEK293 cell homogenates in comparison with brain homogenates. Although PKCζ was present, we found that HEK293 cells did not express PKMζ (although both are expressed in the brain; Fig. 1d), confirming our hypothesis that the action of ZIP is independent of PKMζ. We also confirmed the absence of PKCι/λ, another reported target of ZIP[13], in HEK293 lysates, ruling out its involvement (Fig. 1d). Finally, to investigate whether PKCζ itself was involved in ZIP-mediated GluA1 redistribution, we treated GFP-tagged GluA1-expressing cells with ZIP (5 μM) or vehicle for 40 min and tested for the presence of the Thr410-phosphorylated (activated) PKCζ[14] we found no change in PKCζ phosphorylation state (Fig. 1d).

To analyze ZIP-mediated effects on endogenous AMPARs, we prepared primary cortical neurons, incubated with ZIP and stained with antibodies recognizing the extracellular domain of GluA1, therefore detecting only surface receptors[15]. As shown in Fig. 1e, the fluorescent signal of GluA1 decreased significantly following ZIP treatment. Together, these data suggest that in HEK293 cells and in neurons, ZIP induces redistribution of GluA1 to inner cell compartments and that this action is independent of PKMζ.

**ZIP causes a decrease in AMPAR-EPSCs in the NAc**. Next, we examined whether ZIP-mediated GluA1 redistribution has functional consequences on AMPARs in the brain. We focused on the NAc, since in our previous work we demonstrated the ability of ZIP to interfere with cocaine-conditioned reward in this area[12]. AMPAR-EPSCs in NAc slices of naïve rats were markedly depressed (~50%) following application of ZIP (5 μM; Fig. 1f). The effect of ZIP started ~20 min post application, a delay also observed previously[11]. We obtained similar results with the application of scrZIP (Fig. 1f). Since previous studies suggested that ZIP might affect the viability of the neurons or synaptic membrane integrity[16], thereby causing the apparent reduction in AMPAR EPSCs, we tested the resting membrane potential in the presence of ZIP (Fig. 1g). We observed a stable potential value that started to gradually decline only after 50 min. This indicates that the reduction in AMPAR-EPSCs in the presence of ZIP is not due to cell toxicity (at least not during the first 50 min).

**ZIP/scrZIP act as arginine donors and decrease AMPAR-EPSCs**. As both ZIP and scrZIP elicited similar effects on recombinant and native AMPARs, it appeared that the amino acids of the peptides themselves were responsible for these effects. Of its 13 amino acids, ZIP contains five arginine residues (Fig. 2a). Arginine is a known substrate of nitric oxide synthase (NOS) which produces nitric oxide (NO), and NO-mediated S-nitrosylation of GluA1 was shown to promote AMPARs endocytosis[17]. We therefore hypothesized that cleavage of ZIP by cell peptidases would release free arginine residues, resulting in upregulation of NO production, removal of AMPARs from the cell membrane and thus loss of function. To begin to test this hypothesis, we examined the effect of free arginine on AMPAR-EPSCs. Bath application of L-Arg (25 μM) decreased AMPAR-EPSCs in NAc slices (Fig. 2b), similar to ZIP (Fig. 1f). Next, to investigate whether the free arginine residues were indeed originating from the cleavage of ZIP, we designed two peptides: (1) peptide B, having all, but the five ZIP arginine residues replaced with alanine, and (2) peptide C, having the five ZIP arginine residues, but with the peptidase-cleavage sites removed by appropriate amino acid replacements (Fig. 2c). We then tested the ability of these two new peptides to alter AMPAR-EPSCs in NAc slices. As shown in Fig. 2c (right), the arginine–alanine peptide B (5 μM) inhibited AMPAR-EPSCs to the same extent as ZIP (Fig. 1f) and caused a similar redistribution of GluA1 in HEK293 cells (Fig. 2d). In contrast, the peptidase-resistant peptide C (5 μM), had no effect on AMPAR-EPSCs or on GluA1 localization in HEK293 cells (Fig. 2d). These results suggest that increased levels of free arginine residues, originating from the cleavage of ZIP, mediate the effects of ZIP seen in HEK293 cells and in the NAc.

**Membrane-localized ZIP activates NOS**. Due to the presence of the myristoyl group within the peptide, we hypothesized that ZIP localized to the cell membrane. To test its localization, we tagged ZIP with the fluorescence marker FITC positioned immediately after the myristoyl group (FITC-ZIP). We first confirmed that FITC-ZIP does not induce excitotoxic cell death by performing in vivo microinjections of peptide or vehicle into the NAc. Brain

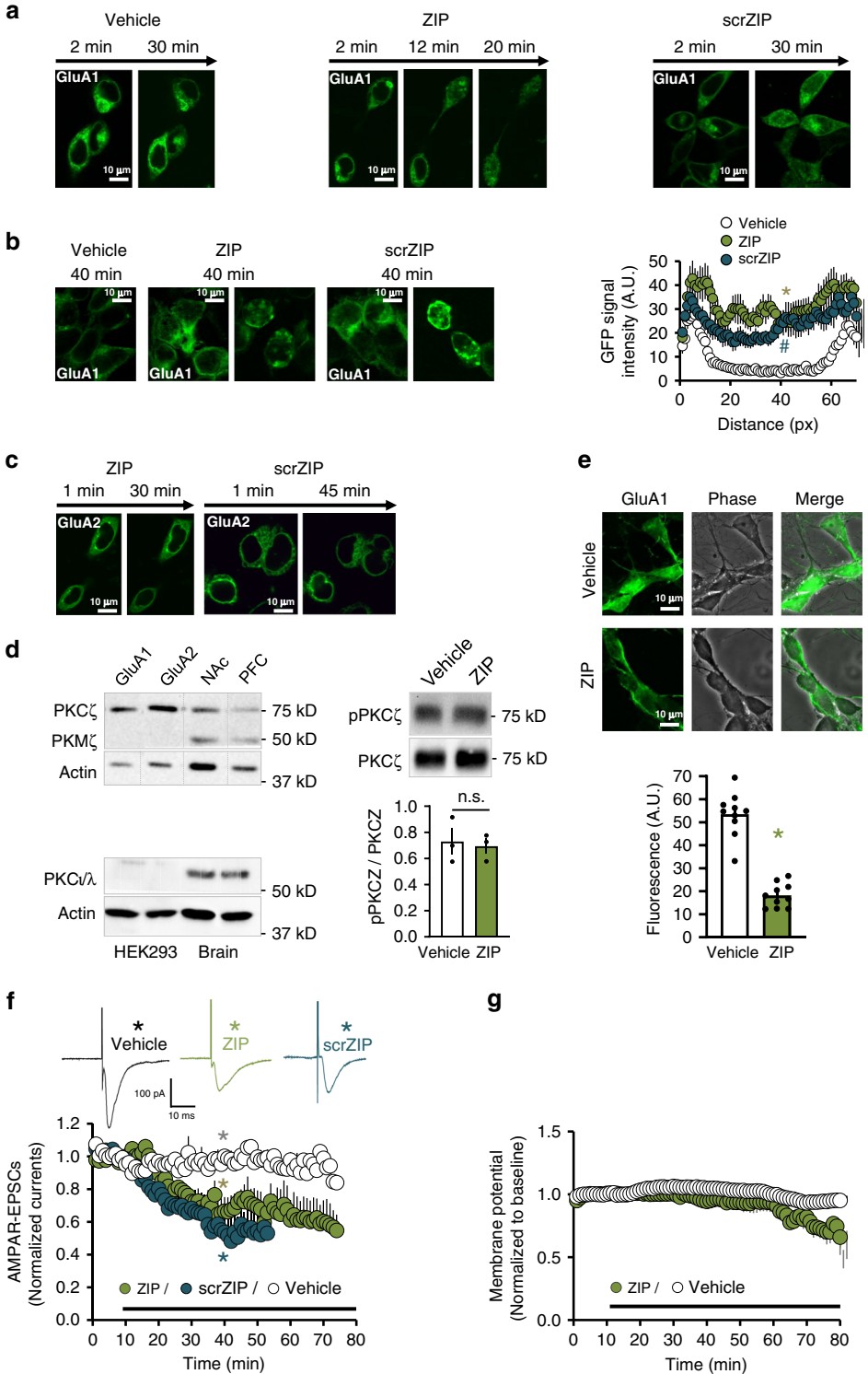

slices from the injected regions prepared 6, 12, and 24 h after the injection, showed that both FITC-ZIP and vehicle injections caused tissue damage during penetration, but without a significant decrease in cell number compared to non-injected animals (Fig. 3a, 24 h post injection). In order to verify that the addition of FITC did not impair ZIP activity, we tested its effect on sustained LTP induced in NAc slices as previously shown for ZIP in the CA1[11]. We first induced LTP by high frequency stimulation (HFS) and following the establishment of LTP we bath applied FITC-ZIP (5 μM). Similar to ZIP[11], FITC-ZIP inhibited the increase of fEPSPs (Fig. 3b).

Next, we incubated FITC-ZIP with HEK293 cells or with primary cortical neurons for 1 h and observed that FITC-ZIP localized to the cell membrane in both cell types (Fig. 3c, top). We hypothesized that once ZIP enters the cell, it will be cleaved by peptidases, resulting in an increased level of free arginine within the cell. To test this hypothesis, we prepared a peptide in which the FITC group was added with a linker between the last two amino acids of ZIP (ZIP-FITC). If ZIP is cleaved within the cell, we than expected the FITC signal to spread throughout the cell. While FITC-ZIP was localized at the membrane, the ZIP-FITC signal was spread throughout the cell in both HEK293 cells and in

**Fig. 1 ZIP causes redistribution of GluA1 in HEK293 cells and primary cortical neurons and decreases AMPAR-EPSCs in NAc neurons. a–c** ZIP scrZIP (5 μM), or vehicle (PBS) were bath applied to HEK293 cells expressing GFP-GluA1 (**a**, **b**) or GFP-GluA2 (**c**). **a** Live cells were imaged using fluorescent confocal microscopy and representative fluorescent images are shown at time points during the incubation, as indicated: (GluA1: ZIP $n = 12$, scrZIP $n = 7$, PBS $n = 3$; GluA2: ZIP $n = 5$, scrZIP $n = 2$). **b** GluA1-expressing cells were fixed after 40 min of incubation with ZIP or vehicle, and the fluorescence was visualized. Graph depicts line scan of GFP signal intensity across the cell. Data expressed as mean ± SEM, arbitrary units (A.U.) with $n = 12$ for ZIP (green), $n = 10$ for scrZIP (blue), $n = 7$ for vehicle (white). Data analyzed by one-way ANOVA ($F = 13.27$, $p < 0.0001$) followed by Tukey post hoc test for multiple comparisons, where vehicle vs. ZIP *$p < 0.0001$, vehicle vs. scrZIP, #$p = 0.0022$. **c** GluA2-expressing cells were live monitored as in (**a**) (GluA2: ZIP $n = 5$, scrZIP $n = 2$). **d** Western blots of HEK293 cell homogenates containing GFP-GluA1 treated with ZIP (5 μM) or vehicle (PBS) and stained with antibodies against activated PKCζ (p-PKCζ) and total PKCζ. Graph depicts the levels of p-PKMζ normalized to total PKMζ ± SEM. Two-tailed unpaired $t$ test revealed no significant difference ($p = 0.7722$, $n = 3$ for each group). **e** ZIP (5 μM) or vehicle (PBS) were bath applied to primary cortical neurons (DIV7) for 40 min. Membrane expression of AMPARs was visualized (left) and quantified with extracellular GluA1 AMPAR antibody (ZIP $n = 10$, vehicle $n = 10$; *$p < 0.0001$, two-tailed $t$ test, data expressed as mean ± SEM. **f** AMPAR-EPSCs from NAc slices evoked by afferent stimulation recorded at holding potential of −70 mV, before and after bath application of ZIP (5 μM) (green; $N = 7$, $n = 7$), scrambled ZIP (5 μM) (blue; $N = 5$, $n = 7$) or vehicle (white; $N = 7$, $n = 7$). Representative EPSC traces taken at the time points indicated by asterisks are shown above. Data presented using normalized values ± SEM. **g** Measurements of membrane resting potential of NAc medium spiny GABAergic neurons following bath application of ZIP or vehicle using current–clamp mode ($N = 5$, $n = 5$). Data presented using normalized values ± SEM.

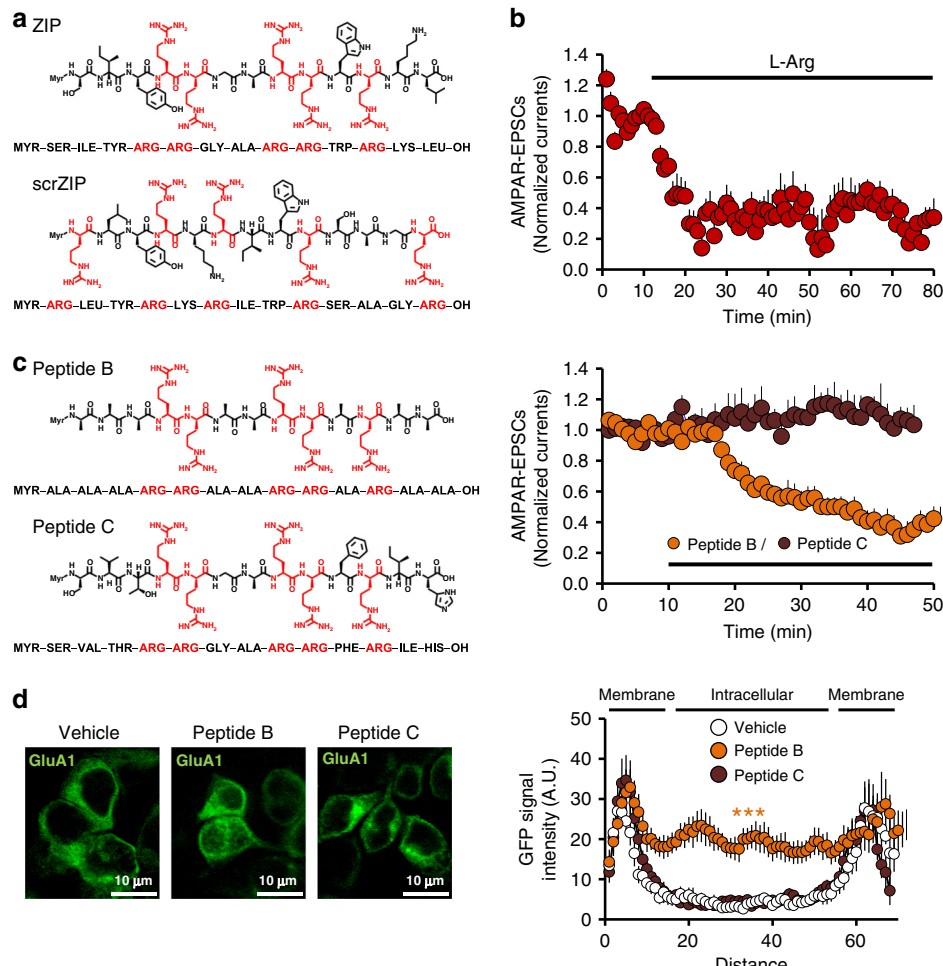

**Fig. 2 ZIP mediates its effects on recombinant and native AMPARs by serving as an arginine donor. a** Peptide structures of ZIP and scr-ZIP. Arginine residues are marked in red. scrZIP contains the ZIP amino acids in a random order. **b** AMPAR-EPSCs, recorded before and after bath application of L-Arg (5 μM, $N = 4$, $n = 4$). Data presented using normalized values ± SEM. **c** Peptide B but not peptide C mimics ZIP effects. Peptide B contains the five arginine residues of ZIP, replacing the eight other amino acids with alanine. Peptide C also contains five arginine residues but lacks its peptidase-cleavage sites due to appropriate replacements (see structures and sequences). Right, AMPAR-EPSCs recorded before and after application of peptide B ($N = 3$, $n = 4$) or peptide C ($N = 3$, $n = 3$). Data presented using normalized values ± SEM. **d** Representative fluorescent images of fixed GFP-GluA1-expressing HEK293 cells, incubated for 40 min prior to the fixation with one of the peptides or vehicle. Graph depicts line scan of GFP signal intensity mean ± SEM (vehicle $n = 7$, Peptide B $n = 10$, Peptide C $n = 9$; $F = 23.85$, $p < 0.0001$, one-way ANOVA, followed by Tukey post hoc test for multiple comparisons, Peptide B vs. vehicle ***$p < 0.0001$, Peptide B vs. Peptide C ***$p < 0.0001$).

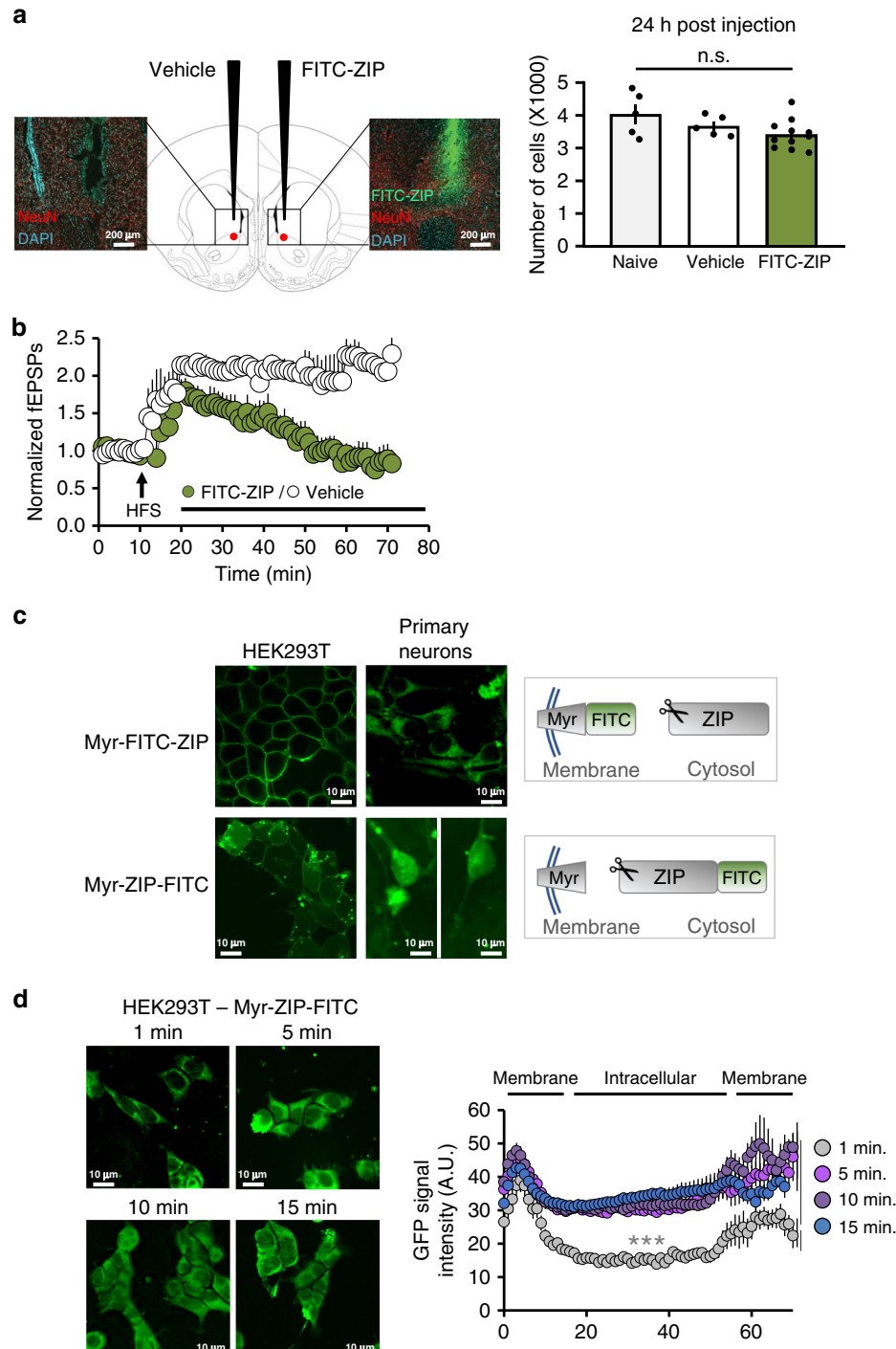

**Fig. 3 ZIP is localized to the membrane in both HEK293 cells and primary cortical neurons. a** Assessment of brain cell viability. FITC-ZIP (green, 1 mM) or vehicle were microinjected into the NAc of rats (AP + 1.75, ML ± 1.48, DV −6.2). Twenty-four hour after injection, animals were sacrificed and brain slices containing the NAc were stained with NeuN (red) or DAPI (blue). Graph depicts the number of DAPI-stained cells mean ± SEM in non-injected (N = 5), vehicle injected (N = 5) and FITC-ZIP injected (N = 11) brains. No significant difference was observed between groups (F = 2.912, p = 0.0802, one-way ANOVA; naïve vs. vehicle p = 0.4740, Naïve vs. FITC-ZIP p = 0.0675, vehicle vs. FITC-ZIP p = 0.5843). **b** Measurements of FITC-ZIP activity and cellular-membrane localization. fEPSPs in NAc slices were monitored before and after high frequency stimulation (HFS; arrow) to induce LTP. FITC-ZIP (5 μM) (green circles; N = 3, n = 4) or vehicle (white circles; N = 3, n = 3) were bath applied 10 min following application of HFS. Data presented using normalized values ± SEM. **c** Representative fluorescent images visualized with confocal microscopy taken from HEK293 cells (left) or mouse embryonic primary cortical neurons (right; DIV7) incubated with FITC-ZIP (5 μM, top; HEK293 N = 5, Neurons N = 5) or ZIP-FITC (5 μM, bottom; HEK293 N = 4, Neurons N = 5) for 1 h. **d** HEK293 cells were incubated with ZIP-FITC (5 μM) for the time indicated in the panel. Fluorescence was visualized with a confocal microscope. Graph depicts line scan of FITC signal intensity mean ± SEM (1 min, n = 13; 5 min, n = 12; 10 min, n = 11; 15 min, n = 12; F = 32.16, p < 0.0001, one-way ANOVA, followed by Tukey post hoc test for multiple comparisons, zero point vs. 5 min. ***p < 0.0001, zero point vs. 10 min. ***p < 0.0001, zero point vs. 15 min. ***p < 0.0001).

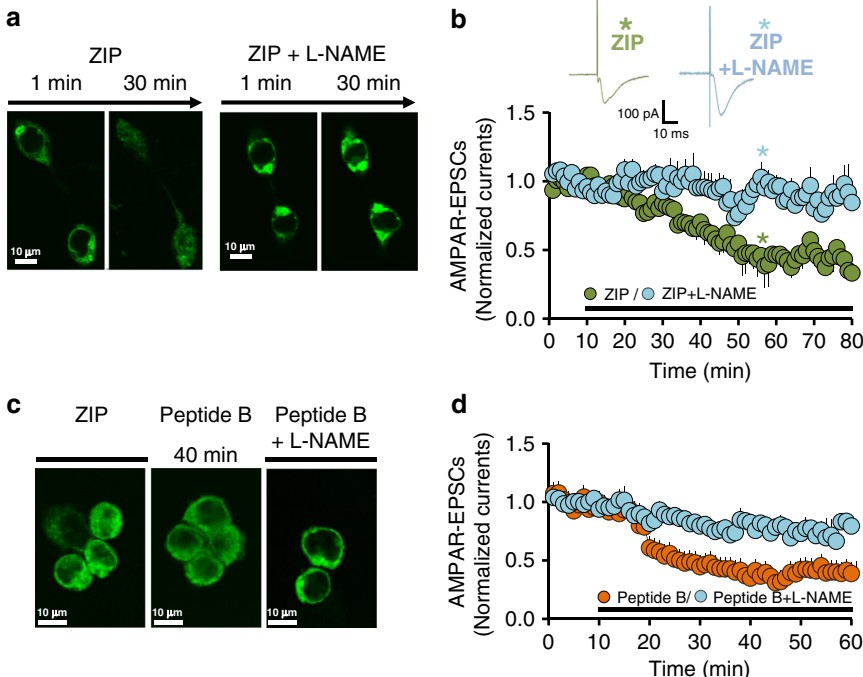

**Fig. 4 ZIP-induced GluA1 redistribution in HEK293 cells and AMPAR-EPSCs decrease in the NAc are mediated by NO. a** Representative fluorescent images taken from continuous live monitoring of HEK293 cells expressing GFP-GluA1, at 1 and 30 min after bath application of ZIP (5 μM), without (left panels, $N = 5$) or with L-NAME (50 μM, right panels, $N = 5$). **b** AMPAR-EPSCs before and after application of ZIP without (green circles; $N = 7$, $n = 7$) or with L-NAME (light blue circles; $N = 7$ $n = 7$). Representative EPSC traces taken at the time indicated by the asterisks are shown on the top. Data presented using normalized values ± SEM. **c** Representative fluorescent images taken from HEK293 cells expressing GFP-GluA1 fixed after 40 min incubation with ZIP (5 μM, left; $N = 5$), Peptide B (5 μM, middle; $N = 6$) and Peptide B (5 μM) + L-NAME (50 μM, right; $N = 5$). **d** AMPAR-EPSCs recorded before and after application of Peptide B in the presence (light blue circles; $N = 5$, $n = 5$) or absence of L-NAME (orange circles; $N = 5$ $n = 5$). Data presented using normalized values ± SEM.

primary cortical neurons (Fig. 3c, bottom). If ZIP acts to stimulate NOS through provision of its substrate L-arginine, ZIP should be rapidly proteolyzed. We therefore treated HEK cells with ZIP-FITC and imaged fluorescence over a short-time period (Fig. 3d). As soon as 5 min after application, the fluorescent signal was spread throughout the cell. Taken together, these results indicate that ZIP undergo a rapid cleavage and strengthens our hypothesis that ZIP acts locally to affect NOS and AMPARs.

**NO inhibition blocks ZIP-mediated effects in HEK293 and NAc.** Since we observed that ZIP could serve as an arginine donor, a known substrate of NOS, we next tested the ability of the NOS inhibitor L-NAME to attenuate ZIP-mediated effects on recombinant GluA1 and native AMPARs. Application of L-NAME together with ZIP inhibited the redistribution of GluA1 in HEK293 cells (Fig. 4a) and completely prevented the reduction of AMPAR-EPSCs by ZIP in NAc slices (Fig. 4b). To confirm that the mechanism of action was indeed through ZIP-donated arginine, we repeated these experiments with the arginine–alanine peptide B. L-NAME prevented both GluA1 redistribution in HEK293 cells (Fig. 4c) and the reduction of AMPAR-EPSCs by peptide B in NAc slices (Fig. 4d), mirroring the effects elicited by ZIP.

Taken together, these results suggest that cleavage of membrane-bound ZIP produces high levels of free arginine residues which activate NOS to produce NO leading, in turn, to the cellular redistribution of GluA1 and reduction of AMPAR-EPSCs.

**ZIP induces S-nitrosylation and phosphorylation of GluA1.** S-nitrosylation of GluA1 at C875 enhances S831 phosphorylation,

facilitates the associated increase in AMPAR conductance and results in AMPAR endocytosis by increasing receptor binding to the AP2 protein of the endocytic machinery[17]. Therefore, we tested whether the ZIP-induced redistribution of GluA1 is mediated via the same signaling pathway, by using a mutant GluA1 in which the cysteine on position 875 was exchanged to serine (GluA1-C875S)[17]. In contrast to wild-type GluA1, application of ZIP or the cleavable arginine-alanine peptide B to HEK293 cells expressing the GluA1-C875S mutant did not affect receptor localization (Fig. 5a, middle row), suggesting that ZIP-mediated AMPAR redistribution requires nitrosylation of GluA1.

As nitrosylation of GluA1 leads to phosphorylation of GluA1-S831[17], we next tested whether ZIP affects the phosphorylation state of GluA1-S831 in NAc neurons. We found that bath application of ZIP or scrZIP (5 μM, 1 h) to NAc-containing brain slices, enhanced GluA1-S831 phosphorylation compared with vehicle-treated slices (Fig. 5b). Given the increased phosphorylation observed following ZIP application, we therefore predicted that mutation of GluA1 S831 will also result in blocking ZIP-induced GluA1 redistribution. We therefore repeated the experiment with ZIP and peptide B using HEK293 cells expressing a non-phosphorylatable GluA1-S831A mutant. Both peptides failed to cause the redistribution of the mutated GluA1 into the cell (Fig. 5a, bottom row). Together, these results strongly suggest that nitrosylation followed by phosphorylation of GluA1 are required for GluA1 redistribution and provide the linkage between nitrosylation and phosphorylation. Altogether, these results support the possibility that ZIP reduces AMPAR-mediated currents through S-nitrosylation and GluA1 phosphorylation that promotes AMPAR endocytosis.

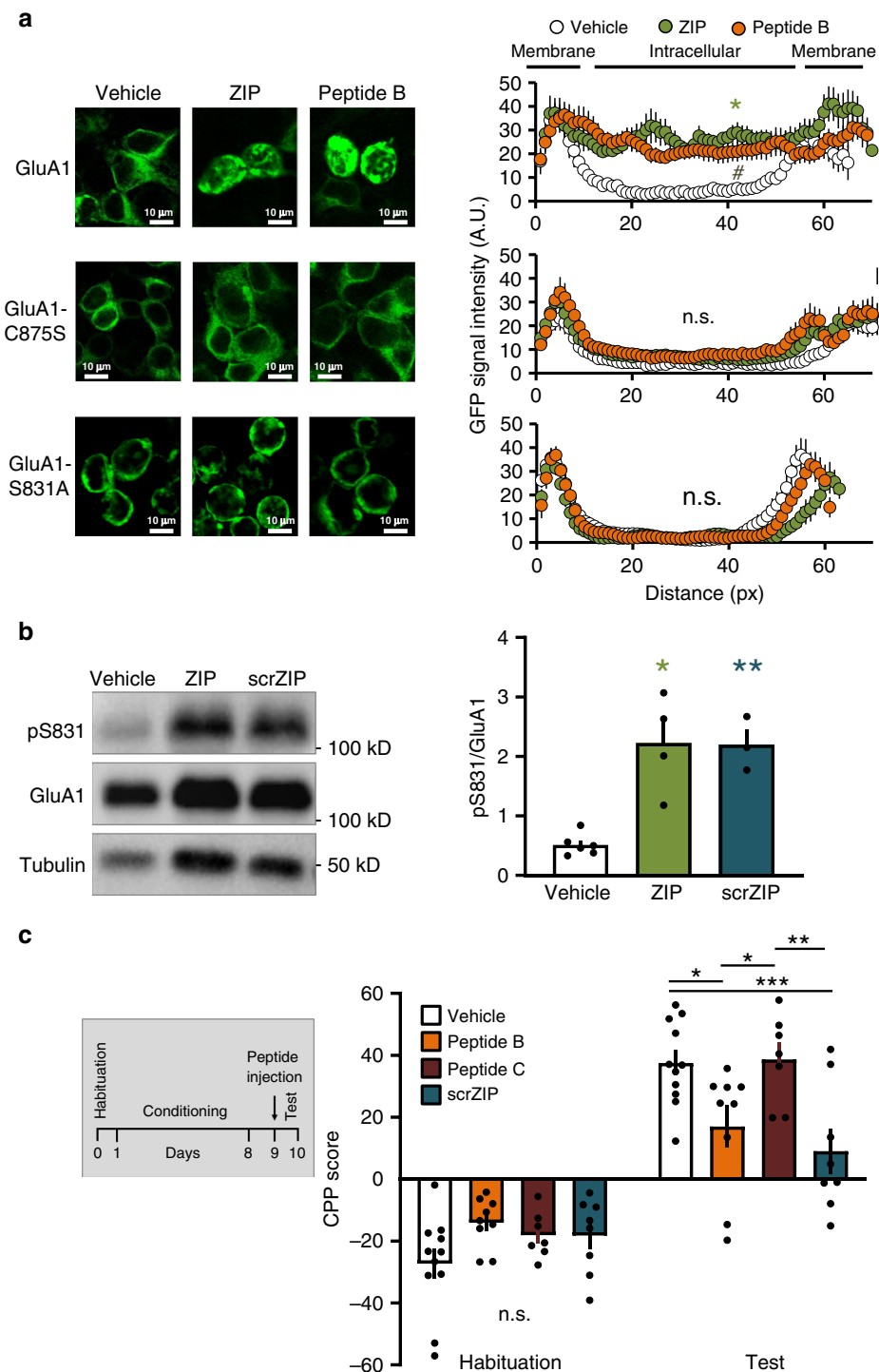

**ZIP, scrZIP, or peptide B decrease the expression of CPP**. We previously reported that microinjection of ZIP into the NAc following conditioning to cocaine abolished the expression of CPP[12]. If the action of ZIP is indeed mediated via an increase in free arginine residues, we predicted that the arginine–alanine peptide B or scrambled ZIP would similarly affect cocaine CPP. A day after the completion of the conditioning paradigm, rats were microinjected with peptide B or scrambled ZIP into the NAc and after 24 h, they were tested for their preference. Rats receiving either peptide showed a significant reduction in the expression of CPP compared with vehicle-injected animals (Fig. 5c). However,

the non-cleavable peptide C, which did not alter GluA1 redistribution or AMPAR-mediated currents in the NAc, had no effect on cocaine CPP (Fig. 5c). These results strongly suggest that the action of ZIP in attenuating cocaine-conditioned reward are mediated by changes in the levels of free arginine in the synapses of NAc neurons.

Together, our findings suggest a molecular mechanism for the action of ZIP. Myristoylated ZIP localizes to the cell membrane and is cleaved by peptidases to produce free arginine residues. Free arginine induces the production of NO by activation of NOS. NO, in turn, nitrosylates (SNO) cysteine 875 on the cytoplasmic

**Fig. 5 ZIP induces S-nitrosylation of GluA1 and increases phosphorylation of S831, and peptide B or scrambled ZIP attenuated the expression of cocaine-conditioned reward. a** Representative fluorescent images of fixed HEK293 cells expressing GFP-GluA1 (upper images), nitrosylation site mutant GluA1-C875S (middle images) or phosphorylation site mutant GluA1-S831A (lower images), incubated for 40 min with vehicle, ZIP or peptide B. Graphs depict line scans of GFP signal intensity mean ± SEM (Upper: WT-GluA1, ZIP $n = 10$, Peptide B $n = 10$, vehicle: $n = 7$). Data analyzed by one-way ANOVA ($F = 10.27$, $p = 0.0006$); Tukey post hoc test for multiple comparison show vehicle vs. ZIP *$p = 0.0005$, vehicle vs. Peptide B #$p = 0.0061$; Middle: GluA1-C875S, vehicle $n = 10$, ZIP $n = 12$, Peptide B $n = 10$; n.s. no significant difference ($F = 2.443$, $p = 0.1046$, one-way ANOVA); Lower: GluA1-S831A, vehicle: $n = 13$, ZIP $n = 11$, Peptide B $n = 12$; no significant difference was observed between groups, $F = 0.4282$, $p = 0.6552$, one-way ANOVA). **b** Western-blot analysis of NAc slice-homogenates incubated prior to the homogenization for 30 min with control vehicle, ZIP or scrZIP (5 µM). Blots were stained with antibodies to GluA1-pS831, total GluA1 and tubulin. Graph indicates the mean change in phosphorylation relative to vehicle-treated (±SEM, vehicle $N = 6$, ZIP $N = 4$, scrZIP $N = 3$; $F = 18.04$, $p = 0.0005$, one-way ANOVA followed by Tukey post hoc test shows significant differences vehicle vs. ZIP *$p = 0.001$, vehicle vs. scrZIP **$p = 0.0023$). **c** Effect of peptide B or scrambled ZIP on CPP. Rats were habituated and conditioned to cocaine as described in methods. One day after the conditioning period, rats were microinjected bilaterally into the NAc with peptide B (20 µg per side; $N = 9$), peptide C (20 µg per side; $N = 7$), scrZIP (20 µg per side; $N = 8$) or vehicle (1 µl per side; $N = 11$) and one day later, tested for CPP. The bar histogram depicts the CPP mean ± SEM (as described in "Methods"). Two-way ANOVA ($F = 7.110$, $p = 0.0003$ for interaction) followed by Sidak's test for multiple comparison within Habituation data (n.s) and test data (vehicle vs. Peptide B *$p = 0.0196$, vehicle vs. Peptide C $p > 0.9999$ n.s., vehicle vs. scrZIP ***$p = 0.0007$, Peptide B vs. Peptide C *$p = 0.0327$, Peptide B vs. scrZIP $p = 0.8571$ n.s., Peptide C vs. scrZIP **$p = 0.0018$).

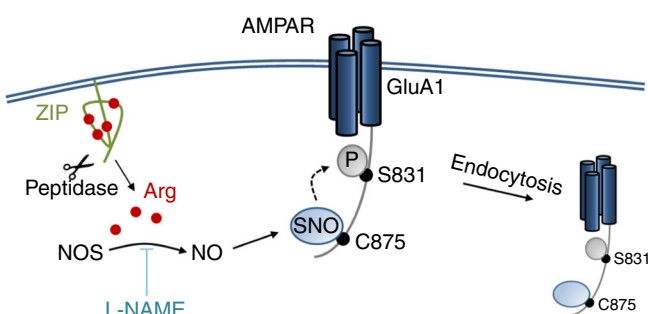

**Fig. 6 Model for the molecular mechanism of ZIP.** Myristoylated ZIP binds to the cell membrane and is digested by peptidases to produce free arginine residues (red circles). Arginine is a substrate of nitric oxide synthase (NOS), which generates nitric oxide (NO). NO in turn, nitrosylates the cysteine residue on position 875 on the cytoplasmic tail of the AMPAR GluA1 subunit, which induces serine phosphorylation of GluA1 on position 831. S831 phosphorylation leads to endocytosis of AMPARs and subsequent functional downregulation.

tail of AMPAR GluA1, which induces phosphorylation (P) of GluA1 on serine 831, leading to endocytosis of the GluA1-containing AMPARs and their downregulation (Fig. 6).

## Discussion

Although ZIP has its origin as a pseudo-substrate of PKMζ, recent studies in PKMζ knock-out mice have confirmed that we do not yet completely understand its mechanism of action. Here, we used HEK293 cells expressing GluA1, primary cortical neurons and electrophysiology in NAc slices to examine the hypothesis that ZIP acts as a regulator of GluA1 activity, independent of PKMζ. We found that ZIP serves as a source of free arginine residues that activate NOS to produce NO, causing nitrosylation of GluA1 and eventually its endocytosis and downregulation. While these results contradict many studies implicating ZIP as a PKMζ inhibitor[1], our data are in line with previous work suggesting that ZIP can act independently of PKMζ to interfere with synaptic plasticity and long-term memory[18,19]. To this end, we tested the ability of ZIP to affect one of the key players in LTP, the AMPARs. Although we have no evidence to show a physical interaction of ZIP with GluA1, our results indicate that ZIP interferes with the normal function of AMPARs. In HEK293 cells expressing GluA1 in the absence of PKMζ, ZIP causes redistribution of GluA1, suggesting that ZIP functionally interacts with GluA1, independent of PKMζ. Therefore, we hypothesized

that the ZIP-induced inhibition of AMPAR-EPSCs in the brain is due to its ability to downregulate the AMPAR activity. Indeed, our findings demonstrate that ZIP decreases AMPAR-EPSCs in NAc slices taken from naïve rats.

As we and others[10,11,16,20] found that ZIP and scrambled ZIP evoke similar outcomes, we speculated that the arginine residues of ZIP themselves contribute to its function. We hypothesized that ZIP binds to the synaptic membrane with the myristoyl group which, following cleavage by peptidases, release free arginine residues thereby increasing the arginine concentration local to the AMPA receptor at the synaptic membrane. Although neurons contain a large amount of intracellular free arginine, they have the capacity to respond to exogenously applied arginine to mediated NO effects, a phenomenon known as "The arginine paradox"[21]. Previous studies have shown that extracellular arginine binds to cell surface receptors activating NOS via G-proteins[22]. Indeed, we found that arginine by itself decreased AMPAR-EPSCs in NAc slices, similar to ZIP. However, we also demonstrated that FITC-tagged peptide cleavage occurred intracellularly and that inhibition of NOS by L-NAME blocked the peptide-induced redistribution of AMPAR-GluA1. Furthermore, intracellular application of L-NAME (via the patch pipette) abrogated the Peptide B (arginine–alanine)-induced reduction in AMPAR-mediated EPSCs in the NAc slices, similarly to ZIP. Although we did not directly measure the intracellular concentration of arginine following peptide application, our results support an effect mediated locally by intracellular arginine. These results are in line with previous studies demonstrating the effect of arginine/NO pathway in regulation of glutamate receptor function[17,23,24].

It has been well documented that AMPAR-GluA1 phosphorylation results in its stabilization, increased conductance and surface expression[25]. In response to NMDA-mediated calcium influx, CaMKII is activated and phosphorylates AMPAR-GluA1 at S831, resulting in increased surface expression and stabilization, underlying early phase LTP. However, among the processes which regulate the phosphorylation of the AMPAR is NO-signaling[26,27], but the specific mechanisms underlying this NO-mediated regulation are only now becoming clear. S-nitrosylation of GluA1 at residue C875 was recently identified as an additional route of AMPAR subunit modification, promoting phosphorylation at the same GluA1-S831 residue. This resulted in subsequent AMPA receptor endocytosis in cortical neurons, thereby regulating synaptic plasticity[17]. Our results indicate that ZIP may act as an arginine donor, hypothesizing that it bypasses the NMDAR and downregulates GluA1 by this pathway. Indeed, we show that the NOS inhibitor L-NAME could block the

ZIP-dependent reduction in EPSCs in the NAc. We also show that the ZIP-induced redistribution of GluA1 in HEK293 cells was not observed for the C875S-mutated GluA1, or in S831A-mutated GluA1. In addition, ZIP or scrZIP induced the phosphorylation of S831 on GluA1 in NAc slices that were treated with ZIP. Most importantly, microinjection of the arginine–alanine peptide B and scrZIP into the NAc following cocaine CPP paradigm, significantly reduced the expression of CPP as we previously reported for ZIP itself[12], while the control peptide C has no effect of the expression of CPP.

Recently several studies raised the possibility that ZIP can act nonspecifically to disrupt normal function of the brain. It has been reported that ZIP as well as its PKMζ-inactive analog, scrZIP, induced a dose-dependent increase in spontaneous activity of neurons in dissociated cultures of rat hippocampal neurons and also caused excitotoxic cell death[16]. In the current study we found that ZIP had no effect on membrane potential during the first 50 min of bath application, whereas its inhibitory effect on AMPAR-EPSCs was apparent. In line with our study, it was shown that in anesthetized rats, ZIP profoundly reduced spontaneous hippocampal local field potentials, comparable in magnitude to infusions of lidocaine, but with a slower onset and longer duration[19]. This suggests that ZIP has no effect on membrane potential, at least not when it interferes with AMPAR activity. Furthermore, we found that when ZIP was injected in vivo to the NAc, at the same concentration used in previous behavioral studies[12], no effect on cell viability was observed 24 h following injection. Therefore, we suggest that the effects of ZIP shown here are not due to its potential toxicity.

Previous studies on PKMζ knockout mice[10,11] showed that scrambled ZIP was able to reverse LTP maintenance in hippocampal slices[11], consistent with other findings that showed partial reversal of fear conditioning memory following scrambled ZIP infusion into the amygdala[20]. In contrast, Shema et al.[8] used scrambled ZIP as a control peptide that did not alter memory in a condition taste aversion paradigm. Indeed, multiple studies support and contradict the role of scrambled ZIP as a control in memory studies[28]. Here, we show that, in addition to peptide B and ZIP[12], scrambled ZIP significantly attenuated the expression of cocaine CPP compared with vehicle-infused animals. These findings support our hypothesis that ZIP, scrambled ZIP and peptide B serve as arginine donors and suggest that vehicle is a more appropriate control when testing the effect of ZIP on learning and memory.

In summary, we propose that the mechanism by which ZIP interferes with memory involves the activation of a NO-signaling pathway that downregulates AMPAR activity and thus leads to long-term memory impairment.

## Methods

**Animals**. The Institutional Animal Care Committee (IACUC) of the Hebrew University (Jerusalem, Israel) approved all procedures. Sprague Dawley male rats (Harlan (Envigo) Laboratories, Jerusalem, Israel), 20–35 days old were used for electrophysiological experiments and for peptide microinjections. Pregnant C57Bl6 mice were used for primary neurons and bred in-house. Animals were housed at an ambient temperature of 22 °C, 30–70% humidity and a 12-h light/dark cycle with the lights on at 7:00 a.m. Food and water were provided ad libitum.

**Peptides**. ZIP: myr-SIYRRGARRWRKL-OH (GLBiochem, China). scrZIP: myr-RLYRKRIWRSAGR-OH (GLBiochem, China), a 13 amino acid peptide composed from the amino acids of ZIP arranged in a random sequence. FITC-ZIP: Fluorescein isothiocyanate (FITC), a fluorescent marker attached to the side chain of extra LYS amino acid that was inserted between myristic acid and first amino acid of ZIP (GLBiochem, China). ZIP-FITC: FITC, attached to the side chain of LYS prior to leucine. Custom-designed peptides: Peptide B: MYR-AAARRAARRARAA-OH (GLBiochem, China), a 13 amino acid peptide containing the ZIP five arginine residues in their original position, but all other amino acids replaced with alanine; Peptide C: MYR-SVTRRGARRFRIH-OH (GLBiochem, China), a 13 amino acid peptide containing the ZIP five arginine residues but the sites of peptidases were destroyed by appropriate amino acid replacements. All peptides were dissolved in

phosphate buffer solution at desired concentrations. L-NAME: Sigma [N5751], L-Arg HCL: National Biochemicals

**Plasmid preparation**. pRK plasmids encoding GFP-tagged GluA1 and GluA2 (both flip isoforms) were a gift from P.H. Seeburg (Max Planck Institute for Medical Research, Heidelberg, Germany)[29]. GFP-tagged GluA1-C875S and GluA1-S831A were prepared using Q5 Site-Directed Mutagenesis Kit (NEB) with following oligonucleotide pairs according to the manufacture recommendation

| | Forward (5′-3′) | Reverse (5′-3′) |
|---|---|---|
| GluA1C875S | ATCCATTCCCTCCATG AGTCACAG | TGCATGGACTT GGGGAAG |
| GluA1 S831A | CCCACAGCAAGCCAT CAATGAAG | ATCAAACAGAAA CCCTTCATCC |

**Cell culture**. Human Embryonic Kidney (HEK)293 cells were passaged weekly and seeded in growth medium (DMEM (−) Calcium (−) Mg (BioInd); 10% fetal bovine serum (HyClone), 1% alanine–glutamine (BioInd), 1% Penicillin–Streptomycin (BioInd), 25 mM HEPES (BioInd)) 24 h before the experiment. HEK293 cells were transfected with AMPARs GluA1, GluA2, GluA1-C875S and GluA1-S831A GFP-tagged plasmids (as appropriate) using transfection reagent JetPrime. Primary neurons were prepared from embryonic 14-day-old (E14) C57/BL6 mice. Cortices were dissected, pooled and roughly chopped. Tissue was incubated in 0.25% trypsin/EDTA, and triturated in DMEM containing 10% FBS. Cells were centrifuged (1500g, 3 min), resuspended and immediately plated at a density of $2 \times 10^5$ cells/cm² on poly-L-lysine/laminin precoated cover slips in Neurobasal growth medium (Neurobasal medium [Gibco], 2% B27 serum free supplement [Thermo], 300 μM glutamine [BioInd], 1% Streptomycin–Amphotericin B). Neurons were maintained in 5% $CO_2$ at 37 °C for 7 days prior the experiment[30].

ZIP (5 μM), scrZIP (5 μM) or vehicle (PBS) were bath applied to culture medium. The localization of the GFP-tagged AMPARs in HEK293 cells was monitored with confocal fluorescent microscopy (FV10, Olympus; with Fluoview v4.2 software) over the time period stated. Alternatively, after 40 min of treatment, cells were fixed in 4% paraformaldehyde solution for 10 min at RT and GFP fluorescence quantified using ImageJ 1.53. For paraformaldehyde-fixed primary neurons, localization of the AMPARs was examined by immunostaining with antibody specific for extracellular domain of GluA1 AMPAR subunit (GluA1-N355/1—Abcam [ab174785]), visualized by confocal fluorescent microscopy (FV10, Olympus; with Fluoview v4.2 software) and quantified as below.

**Quantification of GluA1 redistribution in cells**. Fluorescence intensity was quantified by ImageJ 1.53 (NIH) using the "straight line" tool to manually mark the cell diameter. Fluorescence was measured along the length and expressed as arbitrary units per pixel (A.U.).

**Western blot analysis**. HEK293 cells were detached from the dishes in cold PBS and homogenized in RIPA buffer (320 mM sucrose, 10 mM Tris–HCl (pH 7.4), 1 mM EDTA, 1 mM EGTA, protease inhibitor cocktail (Sigma, P8340), phosphatase inhibitors 1 mM $Na_3VO_4$, 5 mM NaF). Rats were decapitated under isoflurane anesthesia, brains removed and punches of NAc and PFC homogenized in RIPA buffer. Crude homogenates were centrifuged for 10 min at 1000g in 4 °C to separate cell nuclei and unbroken cells. Supernatants were collected and used immediately or stored at −20 °C.

Proteins (20 μg) from brain and HEK293 cell homogenates were resolved by 10% sodium dodecyl sulphate polyacrylamide gel electrophoresis and transferred to a nitrocellulose membrane. Membranes were incubated overnight at 4 °C with appropriate primary antibodies (GluA1—Mercury [AB1504]; GluA1—pS831—Mercury [04-823]; PKMζ—Mercury [07-264]; anti-PKCζ—Abcam [ab59364]; anti-PKCζ-pThr403/410—Cell Signaling [9378]; anti-PKC-iota—Abcam [ab53878]; Actin—Abcam [ab8227]; Tubulin—Sigma [T9026]). Following washes in PBS-T (containing 1% Tween 20), membranes were incubated with appropriate horseradish peroxidase-conjugated secondary antibodies (75 min, RT). Antibody binding was visualized by chemiluminescence (EC-EZL, Biological industries; Clarity Western ECL substrate, BioRad), imaged using BioRad ChemiDoc XRS + apparatus and quantified in BioRad ImageLab software V5.

**Stereotaxic surgery and peptide microinjections**. Prior to the surgery rats were anesthetized with xylazine/ketamine mixture (0.15/0.85) injected IP in 1.1 ml/kg dosage and with tramadol HCL 0.5% solution injected SC in 1.0 ml/kg dosage. Microinjection needle (33 gauge) connected to a 10 μl syringe (Hamilton) was then inserted directly to the NAc and 1 μL of FITC-ZIP, peptide B, peptide C or scrZIP (20 μg prepared in sterile PBS) or PBS alone was injected with a microinjection syringe pump controller (Quintessential Stereotaxic Injector, Stoelting). FITC-ZIP injections were performed unilaterally in NAc shell (coordinates from bregma: AP:

+1.60, ML: ±0.75; −6.00 mm ventral to the skull surface[31]. Following FITC-ZIP injection, rats were set aside for different periods of time (6–24 h after injection), FITC-ZIP expression and post-injection tissue damage were evaluated. Peptide B injections performed bilaterally in NAc shell. Following injection of peptides, rats were returned to the home cage and subjected to CPP experiments.

**Conditioned place preference (CPP).** The CPP paradigm is a standard pre-clinical behavioral model used to study the rewarding or aversive effects of a stimulus. The CPP apparatus includes two chambers of different design and intermediate chamber between them. Animals were divided into experiment and control groups. Following acclimatization, the animals were introduced into the CPP apparatus for a habituation session. At this session, animals were placed in the middle chamber for 5 min and then allowed to explore all chambers for 15 min and the time spent in each side was recorded. The least-preferred compartment for each subject was then assigned to be the drug-paired compartment (biased). Place conditioning: rats received saline (1 ml/kg, IP) or cocaine (15 mg/kg, IP) at alternate days for the next 8 days and placed in the corresponding compartment for 15 min. The test session was arranged in the same way as the habituation sessions (described above). The CPP score calculated for each animal is defined in percents as $\frac{\text{time}_{\text{reward chamber}} - \text{time}_{\text{saline chamber}}}{\text{time}_{\text{reward chamber}} + \text{time}_{\text{saline chamber}}}$.

**Slice preparation and electrophysiological recordings.** Rats were decapitated under isoflurane anesthesia and a brain block containing the NAc was rapidly isolated. Coronal slices, 250 μm thick, were cut using a vibratome VT1000S (Leica, Nussloch, Germany) in ice-cold low-calcium artificial cerebrospinal fluid solution (aCSF) containing (in mM) 119 NaCl, 26 NaHCO$_3$, 11 glucose, 2.5 KCl, 1 NaH$_2$PO$_4$, 2 MgCl$_2$, and 0.5 CaCl$_2$. Slices were transferred to a holding chamber filled with high-calcium (2 mM CaCl$_2$) aCSF solution at 28–30 °C. Slices were allowed to recover for at least one hour before being placed in a recording chamber under continuous flow (2 ml min$^{-1}$) of the aCSF solution. All solutions were saturated constantly with 95% O$_2$ and 5% CO$_2$ throughout the experiments[32]. NAc medium spiny neurons were visualized using an upright microscope with infrared illumination. Whole-cell currents were recorded in a voltage-clamp configuration using a Multiclamp 700B amplifier (Axon Instruments, Foster City, CA) and pClamp 10 acquisition software (Axon Instruments, Foster City, CA). Electrodes pulled from glass capillaries (4-5 MΩ resistance) were filled with an internal solution containing the following (in mM): 120 CsCH$_3$SO$_3$, 20 HEPES, 0.4 EGTA, 2.8 NaCl, 5 TEA-Cl, 2.5 Mg$_2$-ATP, and 0.3 Na$_3$GTP, pH range 7.2–7.4, osmolarity range 275–285 mOsm/L. Recordings of AMPAR-EPSCs were performed at a holding potential of −70 mV. Series and input resistance were monitored continuously with a 4 mV depolarizing step given before every recording sweep and experiments began only after series resistance had stabilized for 5 min at least. Data were discarded if series resistance changed by more than 10% during the recording session. To evoke EPSCs, a bipolar stainless steel stimulating electrode were placed 3–4 mm lateral to the recording electrode and were used to stimulate afferent fibers at a frequency of 0.1 Hz. Data were filtered at 2 kHz, digitized at 10 kHz and collected online. Measurement of membrane potential during bath application of ZIP (Fig. 1g) was performed using current clamp mode.

**Statistical analysis.** All data are presented as mean ± SEM. Electrophysiological results were normalized to baseline or control state. The amplitude of evoked currents and the value of resting membrane potential were calculated by measuring the difference between the peak of the current/potential and the baseline current/potential immediately before the stimulus artifact. The changes in evoked current amplitudes and in resting membrane potential where normalized to the average of the first 10 min. Sample traces shown in the figures represent an average of six traces. For multiple comparisons between groups, one-way or two-way ANOVA was applied as appropriate, followed by Tukey or Sidak post hoc tests (Graphpad Prism v.8). Two-tailed Student's unpaired $t$ test was used for pairwise comparisons. A probability of $p < 0.05$ was considered statistically significant.

**Reporting summary.** Further information on research design is available in the Nature Research Reporting Summary linked to this article.

## Data availability

Source data are provided with this paper. Further data and reagents are available on request from the authors.

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

## Acknowledgements

We thank the late Dr. Jehoshua Katzhendler for his technical support, Dr. Avi Priel for valuable feedback on this manuscript, and Dr. Anat Ben-Yaacov for technical help at the initial stage of the study. This work was supported by the Israel Science Foundation (ISF; 1283/16 R.Y.), the National Institute for Psychobiology in Israel (NIPI), the David R. Bloom Center for Pharmacy at the Hebrew University of Jerusalem (R.Y.), the Royal Society (C.T. and R.Y.) and by an internal grant from IMRIC (Y.S.-B.).

## Author contributions

Conceptualization: A.B. and R.Y.; Methodology: A.B., T.H., C.T., Y.S.-B., and R.Y.; Investigation: A.B., T.H., C.T., Y.S.-B., and R.Y.; Writing—original draft, A.B. and R.Y.; Writing—review and editing: C.T., Y.S.-B., and R.Y.; Funding acquisition: Y.S.-B. and R.Y.

## Competing interests

The authors declare no competing interests.
