## [Peer Review File · Nature Communications]

Reviewers' Comments:

Reviewer #1:

Remarks to the Author:

The association between select atypical PKC isoforms, glutamate receptors, and cognitive performance is not clear due to conflicting reports. The current manuscript provides interesting experimental support that may potentially shed light on discrepancies by identifying a putative noncanonical mechanism for one of the most commonly used reagents for the atypical PKC isoforms PKC/PKMzeta. However, more validation is needed to definitively confirm the authors' interpretations, by inclusion of control experiments to validate mechanism across models, rule out other isoforms, provide rigor, and also the association with the provided cognitive task.

- To definitively indicate that ZIP is acting as an arginine donor, other possibilities need to be eliminated. Specifically, the authors have not ruled out ZIP effects with PKCzeta (which is expressed in examined HEK cells and neuronal tissue based on representative western blots, nor has PKC iota/lambda, which has overlapping sequence homology, also been ruled out as a contributing factor.
- Throughout the manuscript inferences are made to 'direct action' on AMPA Rs by ZIP; however, the no supporting data is provided for such conclusion (e.g., colocalization). Either colocalization of ZIP-FITC with AMPA receptors should be done, or such conclusions need to be revised.
- The evidence is somewhat convincing for GluA1 AMPA receptor regulation in an immortalized cell line, but experimental evidence should be provided that such an event is also occurring in neurons, such as using a primary. Did not test AMPA redistribution in a more natural model (e.g., neuronal culture), given the differences in intracellular signaling cascades between models.
- One of the biggest concerns is with regard to the hypothesis indicating ZIP as an arginine donor and provided data. Specifically, evidence is provided that cleavage for arginine as a novel noncanonical mechanism; at the same time, data is provided that indicates ZIP-FITC undergoes membrane localization. First, nNOS and iNOS (which would be expected to be the predominantly expressed subtypes) is cytosolic in nature compared to eNOS, which has more membrane localization. Second, if indeed ZIP is being subjected to peptidase activity, then it would likely be more difficult for FITC to be localized to the membrane. This discrepancy needs to be addressed.
- In Figure 3, a single graph is provided for number of counted cells to potentially rule out cell death, but based on the methods, counts were taken from samples ranging 6-24 hrs. Given the suggestive decrease following peptide administration, it would be more transparent if the data were temporally displayed so as to evaluate whether or not putative cell death events occur at later time points. Also, based on the provided methods, it is not clear if cell counts – were based only on single or double-labelled nuclei.
- While Peptide B is putatively able to be an arginine donor, evidence is not provided as to whether Peptide C, GluA1 with the mutated s-nitrosylation site would prevent S831 phosphorylation. While the linkage between nitrosylation and phosphorylation site, has been suggested previously by one lab, given the linkage to the hypothesized mechanism, inclusion of such data would increase the rigor and reproducibility of this linkage.
- Similarly, Peptide C was not tested as a putative negative control on cocaine CPP. Further, it's not as though CPP is completely absent (it is present, albeit in a weakened state (depending on specific statistical analysis – not provided, see below). Related, methodology does not clarify the timing of peptide infusions relative to CPP procedure. Depending on whether peptides are delivered during habituation and/or testing will impact interpretation of results (i.e., preventing memory acquisition/consolidation, inhibiting recall).

- Statistical methods are provided, but statistical results are not provided in the manuscript All statistical information should be provided either in the text or in a table and need to include t- and F- values for all t-tests and ANOVAs, respectively, along with main effects where applicable.
- ZIP is not defined in abstract or title
- Figure 3B – representative image for ZIP localization to neuronal membranes should be replaced with a better image, as the current image suggests cytosolic and membrane localization.
- Figure 4B - legend vs labelling is not consistent.
- Peptidase cleavage sites should be identified.
- Number of cells quantified per sample per condition not indicated...
- Coronal map demonstrating cannula targeting not shown.
- Stereotaxic microinjections utilized PBS instead of ACSF, the former which is not as suitable for site-directed delivery to the brain and should be discussed.

Reviewer #2:

Remarks to the Author:

A complete mechanistic explanation of learning and memory has been one of the great research enterprises of neuroscience in our lifetimes. In particular, research of late has adopted a heavy molecular biological approach in order to identify biochemical mechanisms, and indeed, particular intracellular kinases that may serve to make permanent the changes in neural synaptic weights that are thought to underlie learning and memory.

This particular study examines an alternate mechanism for how the supposedly specific PKMzeta inhibitory peptide (ZIP) might induce its LTP- and memory-erasing properties. As promoted by previous researchers, PKMzeta has been touted as “The Memory Molecule” since it is activated by LTP-producing stimuli (as well as behavioural learning), persists throughout expression (of both LTP and memory) and its inactivation by the synthetic peptide ZIP eliminate both LTP and memory when applied directly to brain areas that show (or would be presumed to show) synaptic plasticity. However, a serious problem with this story is that mice with PKMzeta knock-outs still show both LTP memory and, and worse, ZIP still inactivates LTP and memory in these mice (which of course, don't actually express PKMzeta). Despite these contrary findings, there is still continued enthusiasm for the strict PKMzeta hypothesis (e.g., Tsokas et al, 2016;).

Regardless of the steadfast faith shown by some in PKMzeta, alternate hypotheses exist for the memory-abolishing properties of ZIP. In particular, one group has shown that ZIP can promote excitotoxic actions on hippocampal neurons (Sadeh et al, 2015). Another has recently shown that ZIP impairs ongoing synaptic population activity in the hippocampus (LeBlancq et al, 2016). In the present work, Bingor et al show a compelling alternate explanation for the action of ZIP that is highly consistent with most of the prior literature.

Using a cultured cell line that does not express PKMzeta (HEK293), the authors performed live cell fluorescence microscopy to show that ZIP alters the localization of the GLuA1 subunit of the AMPA receptor (tagged with GFP) to the intracellular compartment, as opposed to the membrane surface. Consistent with this, ZIP also depressed AMPA-mediated synaptic currents in neurons from the nucleus accumbens (NAc) as assessed using whole cell techniques in brain slices. Interestingly, the scrambled version of ZIP also had similar influences to ZIP itself.

Given the parallel effects of both ZIP and its scrambled version, the authors hypothesized that a common component of both might yield its effects. Since both are rich in arginine residues, and since arginine is known to activate nitric oxide (shown previously to promote endocytosis of AMPA receptors) they tested arginine itself on AMPA-mediated EPSCs, again in NAc. Like ZIP, arginine depressed excitatory synaptic currents. Interestingly, an alternate manufactured peptide which had the same length as ZIP but had all other non-arginine amino acids replaced with alanine (presumably rendering it inactive with respect to PKMzeta) had the same influence as ZIP itself. Yet another alternate form of ZIP that was rendered resistant to peptidases yielded no such depression, showing that degradation of the peptide with the likely release of arginine was necessary for its action.

Consistent with an effect that would be dependent on NO, the nitric oxide synthase inhibitor L-NAME occluded the effect of ZIP. Furthermore, in HEK293 cells expressing an altered GluA1 that does not allow S-nitrosylation via NO which then leads to increased AMPA receptor endocytosis, ZIP no longer promoted the redistribution of receptors from the cell membrane to the intracellular compartment, and it also reduced the phosphorylation state of GluA1 in NAc neurons which is a necessary step in the endocytosis of AMPA receptors.

Lastly, and most damningly for the ZIP/PKMzeta story, the authors demonstrate that their alanine modified peptide, (like ZIP - as previously shown) reduced a previously acquired cocaine-induced conditioned place preference when infused into the NAc.

The authors' summarize their findings as follows (p8):

"... ZIP localizes to the cell membrane and is cleaved by peptidases to produce free arginine residues. Free arginine induces the production of NO by activation of NOS. NO, in turn, nitrosylates cysteine 875 on the C-tail of AMPAR GluA1, which induces phosphorylation of GluA1 on serine 831, leading to endocytosis of the GluA1-containing AMPAR and its downregulation."

This leads to their major conclusion (p9) that "ZIP acts as a regulator of GluA1 activity, independent of PKMzeta."

Altogether this is a well thought-out and well written piece of work. The authors are to be congratulated for a principled and well purposed stand for following up on previous puzzling findings regarding the actions of ZIP. Despite my complete enthusiasm for this work, the authors may wish to consider some potential suggestions and addendums.

Specific Comments:

Although the authors assert that many researchers have found that both ZIP and its scrambled version promote the same results, I would consider that some prior researches would disagree. Some more detailed commentary and comparisons to prior work will likely be necessary here. I also note that in the authors' previous work (Shabashov et al, 2012) that scrambled ZIP was not used as a control. Would it, like the manufactured peptide and ZIP itself, be expected to diminish CPP as well?

Perhaps related to the above, one of the striking elements of previous neuro-physiological and behavioural studies with ZIP is that testing tends to occur after a period of two hours following application. Although the in vivo experiments here appear to have a delay of one day interposed between infusions and testing, I would be curious to know if the effects that the authors are reporting in both culture and brain slice would be expected to continue past two hours?

Reviewer #3:

Remarks to the Author:

Bingor et al studied the mechanism of synaptic depression and memory erasure by ZIP. The authors tested ZIP effect both in vitro and in vivo through confocal imaging and electrophysiology as well as behavioral experiment. They reported that the custom-made peptide composed of arginine and alanine has effect similar to ZIP. The results suggest that ZIP (and other peptides having arginine) provides arginine by being degraded, which is a substrate of NO-synthase, thereby activating NO signaling pathway inducing GluA1 endocytosis and memory erasure. Although the study tackles an important issue of ZIP specificity, their experimental evidence is not enough to support their hypothesis and to justify the publication of the manuscript in Nature Communications. Details are as follows.

Major points

1. Direct proof that ZIP is cleaved in the cell is lacking. The myristoyl group of ZIP seems to be acting as a membrane anchor, localizing the molecule to lipid membranes. Indeed the authors observe FITC conjugated near the myristoyl is still concentrated near the membrane even after an

hour (Figure 3B). This could be used to prove the point, by use of another ZIP-FITC with a c-terminal FITC conjugate that should be freed from the myristoyl group and therefore cytosolic after cleavage. Showing that the peptidase-resistant Peptide C-FITC does become cytosolic would further strengthen the authors' case.

2. GluA1-S831 phosphorylation is not typically linked to AMPAR endocytosis. To prove that S831 phosphorylation is relevant in the context of ZIP, the authors will need to test cells or mice with a phosphomutant GluA1.

3. Throughout the paper, the authors showed the change of GFP signal inside the cell to evaluate the GluA1 endocytosis. But without estimating the GFP signal on the membrane, it is hard to conclude that there is GluA1 endocytosis with their result. In addition, they seemed to analyze the GFP signal in the nucleus rather than cytosol. Indeed, it is not feasible to discriminate the cytosol from the plasma membrane in a confocal image. And it is unusual that the endocytosed proteins stay in the nucleus.

4. For the behavioral experiment (Figure 5), they microinjected Peptide B instead of ZIP. As pointed out above, it is unclear whether Peptide B and ZIP share a common mechanism. In addition, it was not tested whether Peptide B exerted its effects via a NO-signaling pathway activated by arginines from Peptide B.

Minor points

1. If the measurement of membrane potential in Figure 1G corresponds to V_m , then the methods section should properly describe how V_m is measured during voltage clamp. If it corresponds to holding current, then the label should be corrected and the authors should explain why the holding current decreases after 50 minutes, whereas Sadeh et al. found that ZIP increased holding current over time.

2. Figure 4B shows that L-NAME enhances ZIP-induced depression, which is obviously not the author's conclusion.

Reviewers' comments:

Reviewer #1

- **To definitively indicate that ZIP is acting as an arginine donor, other possibilities need to be eliminated. Specifically, the authors have not ruled out ZIP effects with PKCzeta (which is expressed in examined HEK cells and neuronal tissue based on representative western blots, nor has PKC iota/lambda, which has overlapping sequence homology, also been ruled out as a contributing factor.**

As suggested by the reviewer, to rule out ZIP effects on PKC ζ we used phospho-specific antibodies for the activated form of PKC ζ . We treated HEK cells with ZIP or vehicle for 40 min. and subsequent protein analyzed by WB with anti-phospho PKC ζ (Thr-410) recognizing the activated form of PKC ζ (Chou MM, et al. Curr Biol. 1998;8(19):1069–1077). In ZIP treated and vehicle treated HEK cells, there were no change in p-PKC ζ , suggesting that PKC ζ is not activated in these cells (new figure 1D). In addition, we performed WB analysis using PKCiota/lambda on HEK cells and brain homogenates. While PKCiota/lambda was present in the brain, HEK cells did not express this kinase (new figure 1D). We therefore concluded that there was no involvement of either kinase in ZIP-mediated effects in HEK293 cells. The new results are included in updated figure 1D and the results section on page 4.

- **Throughout the manuscript inferences are made to 'direct action' on AMPARs by ZIP; however, the no supporting data is provided for such conclusion (e.g., colocalization). Either colocalization of ZIP-FITC with AMPA receptors should be done, or such conclusions need to be revised.**

We agree with the reviewer that no data are provided to show physical and direct interaction of ZIP with GluA1. We therefore revised the text accordingly to state that the interaction is functional (page 4, 5 and 10).

- **The evidence is somewhat convincing for GluA1 AMPA receptor regulation in an immortalized cell line, but experimental evidence should be provided that such an even is also occurring in neurons, such as using a primary Did not test AMPA redistribution in a more natural model (e.g., neuronal culture), given the differences in intracellular signaling cascades between models.**

We thank the reviewer for this comment and we therefore conducted new experiments using primary neurons. Primary cortical neurons (DIV7) were prepared from E14 mouse embryos and stained with anti-GluA1 antibodies that recognize the extracellular domain of GluA1 (Kalashnikova E. et al. Neuron 65, 80–93 (2010)). Under these conditions, anti-GluA1 antibodies stain only the membrane-bound GluA1. Following incubation of ZIP for 40 min (a time-point reflecting the experiments in HEK cells), the fluorescent signal intensity of membrane bound GluA1 dropped significantly suggesting that ZIP causes redistribution of GluA1 intracellularly in neurons. The new results were included in new figure 1E and on page 4.

- **One of the biggest concerns is with regard to the hypothesis indicating ZIP as an arginine donor and provided data. Specifically, evidence is provided that cleavage for arginine as a novel noncanonical mechanism; at the same time, data is provided that indicates ZIP-FITC**

undergoes membrane localization. First, nNOS and iNOS (which would have expected to be the predominantly expressed subtypes) is cytosolic in nature compared to eNOS, which has more membrane localization. Second, If indeed ZIP is being subjected to peptidase activity, then it would likely be more difficult for FITC to be localized to the membrane. This discrepancy needs to be addressed.

We thank the reviewer for this helpful comment and we therefore conducted several experiments to clarify this issue as was also raised by reviewer 3. To test whether ZIP is indeed cleaved within the cell, we prepared a peptide in which the FITC group is added with a linker between the last two amino acids of ZIP ("ZIP-FITC"). If ZIP is cleaved within the cell than we expected the FITC signal to spread throughout the cell. While the original ZIP peptide (containing n-terminal FITC downstream of the myristoyl group, "FITC-ZIP") shows a membrane localization, ZIP-FITC signal was spread throughout the cell, strongly suggesting that ZIP is cleaved intracellularly (new figure 3C). In support of the observations in HEK cells, these results were replicated in primary cortical neurons. These data strongly suggest that ZIP is cleaved intracellularly; supporting our hypothesis that ZIP is a donor of arginine residues. The text was updated accordingly (see page 6-7).

• **In Figure 3, a single graph is provided for number of counted cells to potentially rule out cell death, but based on the methods, counts were taken from samples ranging 6-24 hrs. Given the suggestive decrease following peptide administration, it would be more transparent if the data were temporally displayed so as to evaluate whether or not putative cell death events occur at later time points. Also, based on the provided methods, it is not clear if cell counts – were based only on single or double-labelled nuclei.**

As suggested by the reviewer, we separated the data into early (6 hours) and late (24 hours) cell counts post injection and reanalyzed the data, finding no change in cell number at 24 hours; these counts were based on DAPI staining. The results are shown in new figure 3A, and the text were updated accordingly (page 6).

• **While Peptide B is putatively able to be an arginine donor, evidence is not provided as to whether GluA1 with the mutated s-nitrosylation site would prevent S831 phosphorylation. While the linkage between nitrosylation and phosphorylation site, has been suggested previously by one lab, given the linkage to the hypothesized mechanism, inclusion of such data would increase the rigor and reproducibility of this linkage.**

As suggested by the reviewer, we prepared HEK293 cells transfected with a GluA1 plasmid in which S831 was replaced with alanine (GluA1-S831A). We tested the ability of both ZIP and (arginine-alanine) peptide B to affect the localization of GluA1. Both peptides failed to cause the intracellular redistribution of GluA1 (new figure 4A). These results suggest that phosphorylation of S831 on GluA1 is indeed required for GluA1 redistribution and provide the linkage between nitrosylation and phosphorylation. The text was updated accordingly (page 8, 11).

• **Similarly, Peptide C was not tested as a putative negative control on cocaine CPP. Further, it's not as though CPP is completely absent (it is present, albeit in a weakened state (depending on specific statistical analysis – not provided, see below). Related, methodology**

does not clarify the timing of peptide infusions relative to CPP procedure. Depending on whether peptides are delivered during habituation and/or testing will impact interpretation of results (i.e., preventing memory acquisition/consolidation, inhibiting recall).

We have previously shown that microinjection of ZIP into the NAc one day following conditioning to cocaine abolished the expression of cocaine CPP (Shabashov et al, 2009). The schematic in original figure 4C has been updated to reflect the timing of peptide injection. As suggested by the reviewer we repeated the CPP experiments to include peptide C as a putative negative control of ZIP. In addition, we microinjected scr-ZIP as another positive control, predicted to act like ZIP or peptide B. Microinjection of peptide C into the NAc one day following conditioning to cocaine has no effect on the expression of CPP, similar to vehicle (new figure 4C). However, microinjection of scr-ZIP with the same timing resulted in attenuation of CPP; this was similarly observed for peptide B. These results support our hypothesis that the peptides (ZIP, ScrZIP, Peptide B) that serve as arginine donors can indeed attenuate the expression of CPP. Moreover, these results also suggest that interference with memory occurs in the recall phase one day after the learning event (after the consolidation window is closed). We updated the text to include the new results (page 9) and have improved statistical analyses throughout (see below and improved Figure Legends).

- **Statistical methods are provided, but statistical results are not provided in the manuscript All statistical information should be provided either in the text or in a table and need to include t- and F- values for all t-tests and ANOVAs, respectively, along with main effects where applicable.**

We apologize and have added the description and values of all statistical analysis in the figure legends.

- **ZIP is not defined in abstract or title**

We added definition of ZIP to the abstract and title.

- **Figure 3B – representative image for ZIP localization to neuronal membranes should be replaced with a better image, as the current image suggests cytosolic and membrane localization.**

We updated the figure as discussed earlier; these data comprise new figure 3C.

- **Figure 4B - legend vs labelling is not consistent.**

We updated accordingly.

- **Peptidase cleavage sites should be identified.**

The cleavage of peptides by peptidases is highly dependent on the peptide sequence around the cleavage site, often up to four amino acids before or after the cleaved peptide bond. Each protease has its own unique preference that not always includes recognition of all amino acids. Our peptide design aimed to modify the peptide sequence to limit the release of arginine, and indeed these modifications in Peptide C led to no activity. It is very difficult to name the specific cleavage site as we have not identified the exact peptidase responsible for cleavage of the peptides. It is however suggestive that the peptidase recognizes alanine and arginine in Peptide B and can cleave it, a property ascribed to arginyle peptidase (MEROPS data base <https://www.ebi.ac.uk/merops/c qi-bin/pepsum?id=S01.281;type=P>). It is also suggestive that the peptidase has limited activity for peptides that include phenylalanine or threonine as in Peptide C.

- **Number of cells quantified per sample per condition not indicated.**

As suggested by the reviewer, we included the numbers in all figure legends.

- **Coronal map demonstrating cannula targeting not shown.**

We included new cartoon in figure 3A.

- **Stereotaxic microinjections utilized PBS instead of ACSF, the former which is not as suitable for site-directed delivery to the brain and should be discussed.**

For microinjections we follow the protocol of Cetin et al, (Cetin et al., 2006, Nature Protocols 1(6) 3166-3173).

Reviewer #2

- Although the authors assert that many researchers have found that both ZIP and its scrambled version promote the same results, I would consider that some prior researches would disagree. Some more detailed commentary and comparisons to prior work will likely be necessary here. I also note that in the authors' previous work (Shabashov et al, 2012) that scrambled ZIP was not used as a control. Would it, like the manufactured peptide and ZIP itself, be expected to diminish CPP as well?

We agree with the reviewer with respect to prior studies and we added a paragraph to the discussion that deals with potential controversies (page 12). As suggested by the reviewer we conducted CPP experiment using scrambled ZIP. Scrambled ZIP injected into the NAc following cocaine conditioning, significantly attenuated the expression of CPP, similar to peptide B (having all, but the five ZIP arginine residues, replaced with alanine; new figure 4C). These results strongly support our hypothesis that ZIP serves as an arginine donor. The new results are included (page 9).

- Perhaps related to the above, one of the striking elements of previous neuro-physiological and behavioural studies with ZIP is that testing tends to occur after a period of two hours following application. Although the in vivo experiments here appear to have a delay of one day interposed between infusions and testing, I would be curious to know if the effects that the authors are reporting in both culture and brain slice would be expected to continue past two hours?

As was mentioned above, we have previously showed that microinjection of ZIP to the NAc abolished the expression of cocaine CPP (Shabashov et al 2012). In this study, we also demonstrated that the negative effects of ZIP were evident during extinction and reinstatement, which lasted for 10 days after the first expression test. These results suggest that the effect of ZIP is long lasting, proceeding at least 10 days following cocaine conditioning.

Reviewer #3

- **Direct proof that ZIP is cleaved in the cell is lacking. The myristoyl group of ZIP seems to be acting as a membrane anchor, localizing the molecule to lipid membranes. Indeed the authors observe FITC conjugated near the myristoyl is still concentrated near the membrane even after an hour (Figure 3B). This could be used to prove the point, by use of another ZIP-FITC with a c-terminal FITC conjugate that should be freed from the myristoyl group and therefore cytosolic after cleavage. Showing that the peptidase-resistant Peptide C-FITC does become cytosolic would further strengthen the authors' case.**

We thank the reviewer for this helpful comment and we therefore conducted several experiments to clarify this issue (as also suggested by reviewer 1). To test whether ZIP is indeed cleaved within the cell, we prepared a peptide in which the FITC group is added with a linker between the last two amino acid of ZIP ("ZIP-FITC"). If ZIP is cleaved within the cell than we expected the FITC signal to spread throughout the cell. While the original peptide that contained FITC just next to the myristoyl group of ZIP ("FITC-ZIP") shows a membrane localization, the new peptide signal was spread throughout the cell (new figure 3C). The experiments were also conducted in primary cortical neurons, and the results were similar to HEK cells. These results strongly suggest that ZIP is cleaved while it is in the cell and support our hypothesis that ZIP is an arginine donor. The text was updated accordingly (page 6-7).

- **GluA1-S831 phosphorylation is not typically linked to AMPAR endocytosis. To prove that S831 phosphorylation is relevant in the context of ZIP, the authors will need to test cells or mice with a phosphomutant GluA1.**

As suggested by the reviewer, we prepared HEK cells transfected with a GluA1 plasmid in which S831 was replaced with alanine (GluA1-S831A) and tested the ability of both ZIP and peptide B to affect the localization of GluA1. Both peptides failed to cause the redistribution of GluA1 into the cell (new figure 4A). These results suggest that phosphorylation of S831 on GluA1 is indeed required for GluA1 redistribution and provide the linkage between nitrosylation and phosphorylation. The text was updated accordingly (page 8, 11).

- **Throughout the paper, the authors showed the change of GFP signal inside the cell to evaluate the GluA1 endocytosis. But without estimating the GFP signal on the membrane, it is hard to conclude that there is GluA1 endocytosis with their result. In addition, they seemed to analyze the GFP signal in the nucleus rather than cytosol. Indeed, it is not feasible to discriminate the cytosol from the plasma membrane in a confocal image. And it is unusual that the endocytosed proteins stay in the nucleus.**

As suggested by the reviewer, we reanalyzed the GFP signal in HEK cells to represent the dynamic change in tagged-GluA1 throughout the cell. As shown in updated figures 1, 2&3, the intracellular redistribution of GluA1 is highly significant in ZIP- or peptide B-treated cells, suggesting that these peptides induce relocalization of GluA1. However, it is hard to distinguish the amount of GluA1 on the membrane under these conditions.

- **For the behavioral experiment (Figure 5), they microinjected Peptide B instead of ZIP. As pointed out above, it is unclear whether Peptide B and ZIP share a common mechanism. In**

addition, it was not tested whether Peptide B exerted its effects via a NO-signaling pathway activated by arginines from Peptide B.

As suggested by the reviewer, we conducted CPP experiments using scrambled ZIP and peptide C (as a negative control as suggested by reviewer 1). Scrambled ZIP results in a significant attenuation of the expression of CPP (as was shown for ZIP itself, see Shabashov et al., 2012; new figure 4C). To test whether peptide B exerted its effects via a NO-signaling pathway, we conducted both immunohistochemistry with HEK cells and electrophysiology as performed with ZIP. As shown in updated figure 3 D&F, we incubated HEK cells with peptide B in the absence or presence of NOS inhibitor L-NAME (as was done with ZIP) and found that L-NAME blocked GluA1 redistribution. We also recorded AMPA-EPSCs under the same conditions and found that L-NAME blocked peptide B-induced AMPA-mediated currents in NAc slices (figure 3G). These results correlate with the effect of these peptides on GluA1 redistribution in HEK cells and implies that both peptides have the potential to serve as arginine donors. We updated the text to include these results (page 7).

Minor points

- **If the measurement of membrane potential in Figure 1G corresponds to V_m , then the methods section should properly describe how V_m is measured during voltage clamp. If it corresponds to holding current, then the label should be corrected and the authors should explain why the holding current decreases after 50 minutes, whereas Sadeh et al. found that ZIP increased holding current over time.**

We apologize for not clarifying this issue, but the measurements of membrane potentials were performed during current clamp mode (as indicated in the legend of figure 1G). In the study of Sadeh et al., primary hippocampal neurons were used to analyze changes in membrane potential while, in the current study, NAc slices were used and we therefore relate these differences to the different methodologies. We added a description to the methods section on page 24.

- **Figure 4B shows that L-NAME enhances ZIP-induced depression, which is obviously not the author's conclusion.**

We thank the reviewers and fixed the typo.

Reviewers' Comments:

Reviewer #1:

Remarks to the Author:

The authors have addressed my concerns. only a few minor editing concerns remain:

- 1) Line 77: change AMPAR to AMPARs (same on 82 –first occurrence; 90; 107; 128; 228, 242)
- 2) Sentence starting on line 107 is incomplete.
- 3) Line 228: change 'its' to 'their'

Reviewer #2:

Remarks to the Author:

I am highly impressed by the thorough and extensive revisions that these authors have taken with respect to both my and the other reviewers' comments.

I am very satisfied with the answers and additions regarding my first comment but wanted to revisit my second since it appears to have been misunderstood.

Previous work supporting the PKMzeta story with respect to ZIP had always evaluated data in a window of time at least 2hrs following infusions. Others have reported more rapid (and detrimental) effects less than or equivalent to this time window. My interest was not in time windows greater than this 2 hour period but less than or equal. Can the authors not comment using their data from either the cell culture or slice data? I note that the time frame for the electrophysiological studies goes up to 80 minutes but do they have data beyond? I realize that this may not be feasible due to the techniques and preparations used.

As a perhaps related follow up, I note the section in the discussion as follows (p 13 lines 273-278):

"Recently, it was shown that in anesthetized rats, ZIP profoundly reduced spontaneous hippocampal local field potentials, comparable in magnitude to infusions of lidocaine, but with a slower onset and longer duration (19). However, in the current study we found that ZIP had no effect on membrane potential during the first 50 min of bath application, whereas its inhibitory effect on AMPAR-EPSCs was apparent. This suggests that ZIP has no effect on membrane potential, at least not when it interferes with AMPAR activity."

One thing that the authors should realize is that field potentials in vivo most likely represent synaptic potentials in dipolar arrangements of pyramidal neurons (see for example chapter 2 in Electroencephalography - Niedermeyer and Lopes da Silva). From the data presented here, it seems like the effects are highly consistent with this previous study in both the reduction of excitatory synaptic neurotransmission as well as the time frame. Perhaps the "However" is misplaced in their wording here?

Reviewer #3:

Remarks to the Author:

The authors addressed some of my questions but there remain significant concerns in their MS.

1. Liberation of free arginines from ZIP is a critical factor for their hypothesis. To show this clearly, the authors need to measure the concentration of free arginines after ZIP treatment (i.e., measurement of free arginines in an extract from the cells incubated with ZIP using HPLC). It is known that the concentration of free arginines in cells is around 100 – 200 μ M, indicating that a large amount of free arginines is already present in cells. To stimulate NOS as its substrate, ZIP should be proteolyzed very effectively within a short time (10 minutes).
2. The authors need to show that S831 phosphorylation of GluA1 is required for AMPAR

endocytosis in neurons (not in HEK cells).

3. ZIP is known to act specifically on potentiated synapses but not non-potentiated, naïve synapses. Based on the ZIP effect on NAC synapses (usually larger than 50 % inhibition), ZIP is likely to have some effects on non-potentiated synapses. As shown in one figure in their revised MS, ZIP effects should be tested after LTP induction. In addition, the authors need to show that ZIP does not have any significant effects on naïve synapses as shown in previous studies.

Response to reviewer's comments:

Reviewer 1:

Line 77: change AMPAR to AMPARs (same on 82 –first occurrence; 90; 107; 128; 228, 242)

We corrected AMPAR to AMPARs when required throughout the text.

Sentence starting on line 107 is incomplete.

We apologize and corrected the text accordingly.

Line 228: change 'its' to 'their'

We corrected the text accordingly.

Reviewer 2:

Previous work supporting the PKMzeta story with respect to ZIP had always evaluated data in a window of time at least 2hrs following infusions. Others have reported more rapid (and detrimental) effects less than or equivalent to this time window. My interest was not in time windows greater than this 2 hour period but less than or equal. Can the authors not comment using their data from either the cell culture or slice data? I note that the time frame for the electrophysiological studies goes up to 80 minutes but do they have data beyond? I realize that this may not be feasible due to the techniques and preparations used.

The reviewer raises an interesting point regarding <2hr time frame of ZIP activity. We already showed cell culture experiments up to one hour, and the images showed in the paper represents a steady-state of change in fluorescent signal. Therefore, we wouldn't anticipate any further changes with a longer incubation. Regarding electrophysiological experiments, as the reviewer also notes, holding cells at -70mV for 2hr would be technically challenging.

As a perhaps related follow up, I note the section in the discussion as follows (p 13 lines 273-278):

"Recently, it was shown that in anesthetized rats, ZIP profoundly reduced spontaneous hippocampal local field potentials, comparable in magnitude to infusions of lidocaine, but with a slower onset and longer duration (19). However, in the current study we found that ZIP had no effect on membrane potential during the first 50 min of bath application, whereas its inhibitory effect on AMPAR-EPSCs was apparent. This suggests that ZIP has no effect on membrane potential, at least not when it interferes with AMPAR activity. One thing that the authors should realize is that field potentials in vivo most likely represent synaptic potentials in dipolar arrangements of pyramidal neurons (see for example chapter 2 in Electroencephalography - Niedermeyer and Lopes da Silva). From the data presented here, it seems like the effects are highly consistent with this previous study in both the reduction of excitatory synaptic neurotransmission as well as the time frame. Perhaps the "However" is misplaced in their wording here?"

We thank the reviewer for his comment and we apologize for the misunderstanding. We adjusted the discussion in the manuscript as suggested.

Reviewer 3:

Liberation of free arginines from ZIP is a critical factor for their hypothesis. To show this clearly, the authors need to measure the concentration of free arginines after ZIP treatment (i.e., measurement of free arginines in an extract from the cells incubated with ZIP using HPLC). It is known that the concentration of free arginines in cells is around 100 – 200 μ M, indicating that a large amount of free arginines is already present in cells. To stimulate NOS as its substrate, ZIP should be proteolyzed very effectively within a short time (10 minutes).

*As we describe in the paper, we hypothesized that ZIP binds to the synaptic membrane with the myristoyl group which, following cleavage by peptidases, release free arginine residues thereby increasing the arginine concentration local to the AMPA receptor at the synaptic membrane. As the reviewer highlights, cells already contain a large amount of free arginine, theoretically saturating nitric oxide synthase (NOS) with its substrate. However, exogenously applied L-arginine still results in NO-mediated biological effects, a phenomenon known as “The arginine paradox” (Kurz and Harrison, (1997) J Clin Invest. **99(3)** 369-70). Therefore, measuring the contribution of ZIP relative to a large pool of intracellular arginine will not necessarily indicate that the global change in the arginine concentration mediates the effect of ZIP on NOS and AMPA receptor.*

We agree with the reviewer that to stimulate NOS through provision of its substrate L-arginine, ZIP should be proteolyzed very effectively within a short time (10 minutes). We therefore performed time course experiments in HEK cells over a short time period to measure the fluorescence signal of ZIP-FITC (see figure 3C). As shown in new figure 3, 5, 10, and 15 min following application of ZIP-FITC to HEK cells, the fluorescent signal was spread within the cell in all time points measured. These results indicate that ZIP undergo a rapid cleavage and strengthening our hypothesis that ZIP acts locally to affect NOS and AMPA receptor. We added the new data to the results and figures.

The authors need to show that S831 phosphorylation of GluA1 is required for AMPAR endocytosis in neurons (not in HEK cells).

*Based on our data and that of Selvakumar et al (PNAS, 2013 **110(3)** 1077-82) the key event that we are investigating is the nitrosylation of GluA1. Subsequently we and others have shown that this nitrosylation results in phosphorylation of GluA1 (pS831) and its redistribution. Preventing nitrosylation (with L-NAME or by GluA1-C875S mutation) or downstream phosphorylation (GluA1-S831A) abrogates the redistribution phenotype. We believe that mutation of S831 site in neurons will not add significantly to overall message we are trying to convey, namely that ZIP, scr ZIP and peptide B are arginine donors for the NO-mediated nitrosylation of GluA1.*

ZIP is known to act specifically on potentiated synapses but not non-potentiated, naïve synapses. Based on the ZIP effect on NAC synapses (usually larger than 50 % inhibition), ZIP is likely to have some effects on non-potentiated synapses. As shown in one figure in their revised MS, ZIP effects should be tested after LTP induction. In addition, the authors need to show that ZIP does not have any significant effects on naïve synapses as shown in previous studies.

As far as we are aware, ZIP is capable of acting on naïve synapses. Volk et al, ((2013) Nature **493** 420–423) have shown that, in hippocampal slices, application of ZIP resulted in a decrease of approximately 50% in field EPSPs in non-potentiated (non-tetanized) slices. The same magnitude of decrease was shown following a standard high frequency stimulation (HFS, LTP protocol) to hippocampal slices (see panel taken from Volk et al., 2013). Although they recorded fEPSP from these slices, the majority of this potentiation is due to AMPA receptors. In our study, we found a similar reduction in AMPA receptor mediated EPSCs in non-potentiated NAc slices (Figure 1F). In addition, we apologize if figure 3B is unclear, but we have already tested the ZIP effect after LTP induction following HFS.

Reviewers' Comments:

Reviewer #1:

Remarks to the Author:

The authors have mainly addressed the comments of all reviewers once the following points are addressed.

- 1) It should be briefly described in the discussion whether peptide cleavage is occurring outside or intracellularly.
- 2) In the discussion, provide support for S831 phosphorylation in increased/stable membrane expression, acknowledge that phosphorylation has been associated with internalization, and briefly address.
- 3) Provide the time in homecage after peptide infusion prior to testing.

Reviewer #2:

Remarks to the Author:

I am very satisfied with the authors' responses and edits to my previous comments.

Reviewer #3:

Remarks to the Author:

I agree upon the authors' suggestion that NO production is critical for the ZIP effect, but it is still unclear how ZIP leads to a persistent decrease in AMPAR-mediated synaptic transmission.

1. As we describe in the paper, we hypothesized that ZIP binds to the synaptic membrane with the myristoyl group which, following cleavage by peptidases, release free arginine residues thereby increasing the arginine concentration local to the AMPA receptor at the synaptic membrane. As the reviewer highlights, cells already contain a large amount of free arginine, theoretically saturating nitric oxide synthase (NOS) with its substrate. However, exogenously applied L-arginine still results in NO-mediated biological effects, a phenomenon known as "The arginine paradox" (Kurz and Harrison, (1997) J Clin Invest. 99(3) 369-70). Therefore, measuring the contribution of ZIP relative to a large pool of intracellular arginine will not necessarily indicate that the global change in the arginine concentration mediates the effect of ZIP on NOS and AMPA receptor.

We agree with the reviewer that to stimulate NOS through provision of its substrate L-arginine, ZIP should be proteolyzed very effectively within a short time (10 minutes). We therefore performed time course experiments in HEK cells over a short time period to measure the fluorescence signal of ZIP-FITC (see figure 3C). As shown in new figure 3, 5, 10, and 15 min following application of ZIP-FITC to HEK cells, the fluorescent signal was spread within the cell in all time points measured. These results indicate that ZIP undergoes a rapid cleavage and strengthening our hypothesis that ZIP acts locally to affect NOS and AMPA receptor. We added the new data to the results and figures.

Unfortunately, the most important conclusion from the present MS depends upon the 'arginine paradox'. Evidence has been provided to solve the 'arginine paradox' (in which concentrations of intracellular arginine are high enough to saturate nitric oxide synthase (NOS), but application of extracellular arginine is still effective to stimulate NOS). However, no definitive answer has been provided so far. There are even some findings which can explain the 'arginine paradox' without any contribution of intracellular arginine. Firstly, a previous study suggested the presence of extracellular arginine receptors which are G-protein coupled receptors (Josh et al., 2007); that is, extracellular arginine binds to its cell surface receptors which activates NOS via G-proteins.

Secondly, extracellular arginine has been shown to increase de novo synthesis of nitric oxide synthases (Lee et al., 2003); that is, even with saturation of nitric oxide synthase with intracellular arginine, newly synthesized nitric oxide synthases could increase NO production. In addition, application of arginine ethyl ester, which is converted to arginine via esterase in cell interior, appeared to have no effects on NOS activities (Shin et al., 2011), suggesting that changes in intracellular arginine concentrations may not be critical for nitric oxide production.

In sum, it is premature to conclude that free intracellular arginine, which may be liberated from ZIP, can enhance NOS activities. Furthermore, the author did not determine whether free arginine would be liberated from ZIP. The FITC experiments in the present MS do not necessarily mean that free arginine is liberated from ZIP, but merely suggest that ZIP peptides are cleaved.

Ref)

1. Joshi MS1, Ferguson TB Jr, Johnson FK, Johnson RA, Parthasarathy S, Lancaster JR Jr.

Receptor-mediated activation of nitric oxide synthesis by arginine in endothelial cells. *Proc Natl Acad Sci U S A*. 2007 Jun 12;104(24):9982-7. Epub 2007 May 29.

2. Lee J1, Ryu H, Ferrante RJ, Morris SM Jr, Ratan RR. Translational control of inducible nitric oxide synthase expression by arginine can explain the arginine paradox. *Proc Natl Acad Sci U S A*. 2003 Apr 15;100(8):4843-8. Epub 2003 Mar 24.

3. Shin S1, Mohan S, Fung HL. Intracellular L-arginine concentration does not determine NO production in endothelial cells: implications on the "L-arginine paradox". *Biochem Biophys Res Commun*. 2011 Nov 4;414(4):660-3.

2. Based on our data and that of Selvakumar et al (PNAS, 2013 110(3) 1077-82) the key event that we are investigating is the nitrosylation of GluA1. Subsequently we and others have shown that this nitrosylation results in phosphorylation of GluA1 (pS831) and its redistribution. Preventing nitrosylation (with L-NAME or by GluA1-C875S mutation) or downstream phosphorylation (GluA1-S831A) abrogates the redistribution phenotype. We believe that mutation of S831 site in neurons will not add significantly to overall message we are trying to convey, namely that ZIP, scr ZIP and peptide B are arginine donors for the NO-mediated nitrosylation of GluA1.

HEK cells was used in the previous studies, which the authors mentioned in their rebuttal. Indeed, there is no evidence for involvement of phosphorylation of GluA1 (pS831) in AMPAR internalization in neurons (on the contrary, it is well known that the same phosphorylation is critical for AMPAR externalization in neurons). Please note that HEK cells do not have the postsynaptic density which is a specialized structure for synaptic receptor trafficking. The author should determine whether phosphorylation of GluA1 (pS831) is critical for AMPAR internalization in neurons.

3. As far as we are aware, ZIP is capable of acting on naïve synapses. Volk et al, ((2013) *Nature* 493 420–423) have shown that, in hippocampal slices, application of ZIP resulted in a decrease of approximately 50% in field EPSPs in non-potentiated (non-tetanized) slices. The same magnitude of decrease was shown following a standard high frequency stimulation (HFS, LTP protocol) to hippocampal slices (see panel taken from Volk et al., 2013). Although they recorded fEPSP from these slices, the majority of this potentiation is due to AMPA receptors. In our study, we found a similar reduction in AMPA receptor mediated EPSCs in non-potentiated NAc slices (Figure 1F). In addition, we apologize if figure 3B is unclear, but we have already tested the ZIP effect after LTP induction following HFS.

In the original study by the Sacktor group, ZIP showed no effect on naïve synapses (Figure 1D, Pastalkova et al., 2006). If ZIP decreased excitatory synaptic transmission at naïve synapses, it would be no better than AMPAR inhibitors such as NBQX and ketamine. The example, which was shown in their rebuttal, was just an exception. Furthermore, in this study, ZIP effects on naïve synapses started to develop around 40 min after start of ZIP treatment, inconsistent even with the authors' results in which the ZIP effect started to develop within 5 min after start of ZIP treatment.

Ref)

Pastalkova E1, Serrano P, Pinkhasova D, Wallace E, Fenton AA, Sacktor TC. Storage of spatial information by the maintenance mechanism of LTP. *Science*. 2006 Aug 25;313(5790):1141-4.

Reviewer #1:

1) It should be briefly described in the discussion whether peptide cleavage is occurring outside or intracellularly.

In response to Reviewer 1's concern regarding whether peptide cleavage was intra- or extracellular, we have provided a detailed discussion incorporating similar suggestions from Reviewer 3. Our results imply that the effect we observe is mediated locally by intracellular arginine, resulting from the cleavage of the peptides. The full addition to the discussion is as follows:

We hypothesized that ZIP binds to the synaptic membrane with the myristoyl group which, following cleavage by peptidases, release free arginine residues thereby increasing the arginine concentration local to the AMPA receptor at the synaptic membrane. Although neurons contain a large amount of intracellular free arginine, they have the capacity to respond to exogenously applied arginine to mediated NO effects, a phenomenon known as "The arginine paradox" (Kurz and Harrison, 1997). Previous studies have shown that extracellular arginine binds to cell surface receptors activating NOS via G-proteins (Joshi et al., 2007). Indeed, we found that arginine by itself decreased AMPAR-EPSCs in NAc slices, similar to ZIP. However, we also demonstrated that FITC-tagged peptide cleavage occurred intracellularly and that inhibition of NOS by L-NAME blocked the peptide-induced redistribution of AMPAR-GluA1. Furthermore, intracellular application of L-NAME (via the patch pipette) abrogated the Peptide B (arginine-alanine)-induced reduction in AMPAR-mediated EPSCs in the NAc slices, similarly to ZIP. Although we did not directly measure the intracellular concentration of arginine following peptide application, our results support an effect mediated locally by intracellular arginine.

2) In the discussion, provide support for S831 phosphorylation in increased/stable membrane expression, acknowledge that phosphorylation has been association with internalization, and briefly address.

In order to provide a balanced argument concerning S831 phosphorylation as requested by the Reviewers, we have included the following paragraph which acknowledges the contribution of S831 to insertion and stabilization of AMPAR into the membrane. We also highlight the NO-mediated mechanism of AMPAR-GluA1 endocytosis which occurs through phosphorylation of the same site.

It has been well documented that AMPAR-GluA1 phosphorylation results in its stabilization, increased conductance and surface expression (Yokoi et al., 2012). In response to NMDA-mediated calcium influx, CaMKII is activated and phosphorylates AMPAR-GluA1 at S831, resulting in increased surface expression and stabilization, underlying early phase LTP. However, among the processes which regulate the phosphorylation of the AMPAR is NO-signaling (Huang et al., 2005; Ivanova et al., 2020), but the specific mechanisms underlying this NO-mediated regulation are only now becoming clear. S-nitrosylation of GluA1 at residue C875 was recently identified as an additional route of AMPAR subunit modification, promoting phosphorylation at the same GluA1-S831 residue. This resulted in subsequent

AMPA receptor endocytosis in cortical neurons, thereby regulating synaptic plasticity (Selvakumar et al., 2013).

3) Provide the time in homepage after peptide infusion prior to testing.

We have clarified details of the timeline of peptide infusion as part of the CPP experiment. We have added a sentence to the results section and have rearranged the pertinent information contained within the figure legend (Figure 5C) to make it clearer.

Reviewer #2

I am very satisfied with the authors' responses and edits to my previous comments.

Thank you for your help in improving our manuscript

Reviewer #3

I agree upon the authors' suggestion that NO production is critical for the ZIP effect, but it is still unclear how ZIP leads to a persistent decrease in AMPAR-mediated synaptic transmission.

Unfortunately, the most important conclusion from the present MS depends upon the 'arginine paradox'. Evidence has been provided to solve the 'arginine paradox' (in which concentrations of intracellular arginine are high enough to saturate nitric oxide synthase (NOS), but application of extracellular arginine is still effective to stimulate NOS). However, no definitive answer has been provided so far. There are even some findings which can explain the 'arginine paradox' without any contribution of intracellular arginine. Firstly, a previous study suggested the presence of extracellular arginine receptors which are G-protein coupled receptors (Josh et al., 2007); that is, extracellular arginine binds to its cell surface receptors which activates NOS via G-proteins. Secondly, extracellular arginine has been shown to increase de novo synthesis of nitric oxide synthases (Lee et al., 2003); that is, even with saturation of nitric oxide synthase with intracellular arginine, newly synthesized nitric oxide synthases could increase NO production. In addition, application of arginine ethyl ester, which is converted to arginine via esterase in cell interior, appeared to have no effects on NOS activities (Shin et al., 2011), suggesting that changes in intracellular arginine concentrations may not be critical for nitric oxide production.

In sum, it is premature to conclude that free intracellular arginine, which may be liberated from ZIP, can enhance NOS activities. Furthermore, the author did not determine whether free arginine would be liberated from ZIP. The FITC experiments in the present MS do not necessarily mean that free arginine is liberated from ZIP, but merely suggest that ZIP peptides are cleaved.

Reviewer 1 had a similar concern regarding whether peptide cleavage was intra- or extracellular. We have amended our discussion (as shown below) to take account of this important distinction and we are grateful to Reviewer 3 for the references, some of which are now incorporated into the manuscript. Having re-examined our data, we believe our results imply a local and intracellular effect on AMPA receptors, resulting from the cleavage of the peptides and release of arginine. Although our arginine experiment (in which arginine was bath applied to the slice) could argue for a G-protein coupled receptor mechanism (Joshi et al, 2007), our ZIP and peptide B experiments

produce the same effect and we have shown through FITC labelling, that these peptides are cleaved intracellularly. In addition, we find that intracellular application of L-NAME through the patching pipette blocks the peptide-induced reduction in AMPA EPSCs, both for ZIP and the arginine-alanine Peptide B. We describe in the text below:

We hypothesized that ZIP binds to the synaptic membrane with the myristoyl group which, following cleavage by peptidases, release free arginine residues thereby increasing the arginine concentration local to the AMPA receptor at the synaptic membrane. Although neurons contain a large amount of intracellular free arginine, they have the capacity to respond to exogenously applied arginine to mediated NO effects, a phenomenon known as “The arginine paradox” (Kurz and Harrison, 1997). Previous studies have shown that extracellular arginine binds to cell surface receptors activating NOS via G-proteins (Joshi et al., 2007). Indeed, we found that arginine by itself decreased AMPAR-EPSCs in NAc slices, similar to ZIP. However, we also demonstrated that FITC-tagged peptide cleavage occurred intracellularly and that inhibition of NOS by L-NAME blocked the peptide-induced redistribution of AMPAR-GluA1. Furthermore, intracellular application of L-NAME (via the patch pipette) abrogated the Peptide B (arginine-alanine)-induced reduction in AMPAR-mediated EPSCs in the NAc slices, similarly to ZIP. Although we did not directly measure the intracellular concentration of arginine following peptide application, our results support an effect mediated locally by intracellular arginine.

HEK cells was used in the previous studies, which the authors mentioned in their rebuttal. Indeed, there is no evidence for involvement of phosphorylation of GluA1 (pS831) in AMPAR internalization in neurons (on the contrary, it is well known that the same phosphorylation is critical for AMPAR externalization in neurons). Please note that HEK cells do not have the postsynaptic density which is a specialized structure for synaptic receptor trafficking. The author should determine whether phosphorylation of GluA1 (pS831) is critical for AMPAR internalization in neurons.

To address this concern regarding the dual nature of the phosphorylation site at AMPAR-GluA1 S831, we have provided a more stringent review of the literature in our discussion (see below). This acknowledges the contribution of S831 to insertion and stabilization of AMPAR into the membrane. This was demonstrated by Selvakumar and colleagues who showed AMPAR-GluA1 S831 phosphorylation and subsequent AP-2-mediated endocytosis in primary cortical neurons (Selvakumar et al., 2013). We also highlight the NO-mediated mechanism of AMPAR-GluA1 endocytosis which occurs through phosphorylation of the same site.

It has been well documented that AMPAR-GluA1 phosphorylation results in its stabilization, increased conductance and surface expression (Yokoi et al., 2012). In response to NMDA-mediated calcium influx, CaMKII is activated and phosphorylates AMPAR-GluA1 at S831, resulting in increased surface expression and stabilization, underlying early phase LTP. However, among the processes which regulate the phosphorylation of the AMPAR is NO-signaling (Huang et al., 2005; Ivanova et al., 2020), but the specific mechanisms underlying this NO-mediated regulation are only now becoming clear. S-nitrosylation of GluA1 at residue C875 was recently identified as an additional route of AMPAR subunit modification, promoting phosphorylation at the same GluA1-S831 residue. This resulted in subsequent AMPA receptor endocytosis in cortical neurons, thereby regulating synaptic plasticity (Selvakumar et al., 2013).

In the original study by the Sacktor group, ZIP showed no effect on naïve synapses (Figure 1D, Pastalkova et al., 2006). If ZIP decreased excitatory synaptic transmission at naïve synapses, it

would be no better than AMPAR inhibitors such as NBQX and ketamine. The example, which was shown in their rebuttal, was just an exception. Furthermore, in this study, ZIP effects on naïve synapses started to develop around 40 min after start of ZIP treatment, inconsistent even with the authors' results in which the ZIP effect started to develop within 5 min after start of ZIP treatment.

We appreciate the point of view of the reviewer, however, the major aim of our study is to shed light on the molecular mechanisms of ZIP action, and in particular, its stimulation of nitric oxide-mediated signaling. Classical AMPA receptor antagonists such as NBQX act as competitive inhibitors by binding to the ligand-binding domain on the AMPA receptor. Therefore, application of such antagonists results in a complete blockade of the receptor. In contrast, application of ZIP triggers post-translational modification of AMPA-GluA1 (as described by us and Selvakumar *et al.*) that does not fully block the receptor. Rather ZIP act as subtle regulator both in naïve synapses and following LTP/LTD. This is highlighted by its action on GluA1-S831 as discussed earlier; this phosphorylation site has a dual action of endocytosis and stabilizing AMPA receptors on the membrane. We are aware of the variability of onset of action of ZIP in electrophysiological studies and indeed in the Volk paper (2013), changes in the fEPSPs were observed 20 min after bath application of ZIP. Differences between onset of action of ZIP between our data and that of Volk and colleagues could be due to a number of reasons including that we conducted single cell recordings whereas they used field recordings, we measured EPSCs in the NAc and they measured EPSCs from hippocampal slices.